# Selective electrochemical reduction of nitric oxide to hydroxylamine by atomically dispersed iron catalyst

Dong Hyun Kim[1,8], Stefan Ringe [2,8], Haesol Kim [1], Sejun Kim[3], Bupmo Kim [4], Geunsu Bae [1], Hyung-Suk Oh [5], Frédéric Jaouen [6], Wooyul Kim[7✉], Hyungjun Kim [3✉] & Chang Hyuck Choi [1✉]

Electrocatalytic conversion of nitrogen oxides to value-added chemicals is a promising strategy for mitigating the human-caused unbalance of the global nitrogen-cycle, but controlling product selectivity remains a great challenge. Here we show iron–nitrogen-doped carbon as an efficient and durable electrocatalyst for selective nitric oxide reduction into hydroxylamine. Using in operando spectroscopic techniques, the catalytic site is identified as isolated ferrous moieties, at which the rate for hydroxylamine production increases in a super-Nernstian way upon pH decrease. Computational multiscale modelling attributes the origin of unconventional pH dependence to the redox active (non-innocent) property of NO. This makes the rate-limiting NO adsorbate state more sensitive to surface charge which varies with the pH-dependent overpotential. Guided by these fundamental insights, we achieve a Faradaic efficiency of 71% and an unprecedented production rate of 215 $\mu$mol cm$^{-2}$ h$^{-1}$ at a short-circuit mode in a flow-type fuel cell without significant catalytic deactivation over 50 h operation.

[1] School of Materials Science and Engineering, Gwangju Institute of Science and Technology, Gwangju, Republic of Korea. [2] Department of Energy Science and Engineering, Daegu Gyeongbuk Institute of Science and Technology, Daegu, Republic of Korea. [3] Department of Chemistry, Korea Advanced Institute of Science and Technology, Daejeon, Republic of Korea. [4] Department of Chemical Engineering, Pohang University of Science and Technology, Pohang, Republic of Korea. [5] Clean Energy Research Center, Korea Institute of Science and Technology, Seoul, Republic of Korea. [6] ICGM, Université de Montpellier, CNRS, ENSCM, Montpellier, France. [7] Department of Chemical and Biological Engineering, Sookmyung Women's University, Seoul, Republic of Korea. [8]These authors contributed equally: Dong Hyun Kim, Stefan Ringe. ✉email: wkim@sookmyung.ac.kr; linus16@kaist.ac.kr; chchoi@gist.ac.kr

The nitrogen-cycle is vital for sustainability of the terrestrial, marine, and atmospheric ecosystems on Earth, and comprises the key stages of nitrogen fixation-nitrification-denitrification[1]. However, the large-scale intensification of a fertiliser-dependent agriculture and the massive combustion of fossil fuels have significantly unbalanced Nature's nitrogen-cycle[2,3]. The anthropogenic inflow of nitrogen oxides ($NO_x$) leads to its fast accumulation, causing serious environmental and health problems[4,5]. Therefore, the electrocatalytic reduction of $NO_x$ from renewable energy is a promising strategy to bring the nitrogen-cycle back into balance[6], alleviating $NO_x$ accumulation and at the same time producing useful chemicals. In particular, hydroxylamine ($NH_2OH$) is an interesting compound, involved in the production of caprolactam (the base chemicals for the nylon industry) as well as a potential hydrogen-carrier for the renewable energy society[7].

In the series of nitrogen reduction steps starting from nitrate, the catalytic reduction of nitric oxide (NO) is a key step to allow for the further reduction of nitrogen, determining the nature of the further reduced nitrogen products (e.g., $N_2O$, NO, $NH_2OH$, and $NH_3$)[8,9]. Noble metal electrocatalysts such as Pt and Pd typically produce $N_2O/N_2$ (low overpotential region) and $NH_2OH/NH_3$ (high overpotential region) from the NO reduction reaction (NORR)[7,9–11]. Meanwhile, some non-noble organometallic complexes (e.g., metallo-porphyrin/phthalocyanine (Pc) complexes, vitamin $B_{12}$, and Prussian blue) catalyse the NORR primarily to $NH_2OH$ and $NH_3$[7,12–15]. In contrast, heme proteins (e.g., myoglobin and haemoglobin) mainly produce $N_2O$, in spite of their structurally similar active sites (i.e., Fe–$N_4$ core)[16,17]. This astounding difference in NORR selectivity despite similar core active site structure was also observed in biological systems. For instance, enzymatic NO reduction by cytochrome P450nor (single-heme) and nitrite reduction by cytochrome c′ nitrite reductase (multi-heme), despite identical NO–$Fe^{II}N_4$ intermediate adduct structures, lead to $N_2O$ and $NH_3$, respectively[18–20].

Much effort has thus been devoted to identifying the physico-chemical parameters that govern the NORR selectivity on single-site and metallic surfaces[11,21–25]. Improved understanding on the NORR electrocatalysis by heme (iron protoporphyrin IX) has been reached via its controlled immobilisation on a graphite surface[21–23]. Two different NORR pathways were identified: pH-dependent ($NH_2OH$ formation) and pH-independent ($N_2O$ formation) pathways, the selectivity of which is affected by electrolyte pH, NO concentration, and electrode potential. By controlling these experimental parameters, highly selective NO-to-$NH_2OH$ conversion was also achieved with a rotating disk electrode (RDE) setup[21–23]. Along with the fundamental backgrounds, electrochemical $NH_2OH$ synthesis has also been demonstrated at device-level with catalysts incorporating heme-like moieties[7,26–28]. Further progress is however still needed to improve its productivity and to secure operational durability for practical applications. For instance, unlike the broad range pH (2–12) typically applicable for half-cell studies[13,16,29,30], $NH_2OH$ production at device-level (e.g., the $H_2$–NO fuel cells) requires strongly acidic electrolytes (3–5 M, pH < 0) to suppress a competitive $N_2O$ production[7,26–28]. Although rapid catalytic deactivation would be expected due to the dissolution of the coordinated metal ion in such highly corrosive conditions[31–33], catalytic stability for NORR has hitherto been underinvestigated[28]. Therefore, the development of new catalytic materials with high activity, selectivity, and stability is the next challenge for the success of $NH_2OH$ production from the artificial electrochemical denitrification.

Herein, we have studied the NORR electrocatalysis of a single-atom Fe catalyst, in which the heme-like active $FeN_xC_y$ moieties are covalently bonded to the carbonaceous substrate (i.e., Fe–N–C catalyst). Because the $FeN_xC_y$ moieties in this catalyst were shown to not suffer from strong Fe demetalation in the acidic electrolytes[34], this well-defined catalyst has provided a suitable platform for both fundamental understandings and device-level operations of $FeN_xC_y$ moieties under highly corrosive reaction conditions. The NORR selectivity and the nature of catalytic sites have been investigated by advanced ex/in situ analytical approaches combined with computational electrolyte-aware density functional theory (DFT) calculations and micro-kinetic modelling. Finally, we achieved effective and durable $NH_2OH$ production on the single-atom Fe catalyst in a prototypical $H_2$–NO fuel cell reactor.

## Results

**Voltammetry of FeNC-dry-0.5 in NORR**. The catalyst with single-atom Fe sites (labelled 'FeNC-dry-0.5') was synthesised by pyrolysis of $Fe^{II}$ acetate, 1,10-phenanthroline (phen), and $Zn^{II}$ zeolitic imidazolate framework (ZIF-8). The labelling refers to homogenised condition (i.e., ball-milling of 'dry' precursor powders) and Fe content in the precursor mixture before pyrolysis at 1323 K (see details in Methods section). As well-identified in our previous works[35,36], this catalyst is solely composed of isolated $FeN_xC_y$ moieties (total Fe content ca. 1.5 wt%, no discernible Fe particles) conjugated on N-doped carbon substrate, as confirmed by a series of physical characterisation (see details in Supplementary Note 1 and Supplementary Figs. 1–5). Especially, $^{57}Fe$ Mössbauer spectroscopy and Fe K-edge extended X-ray absorption fine structure (EXAFS) reveal only two quadrupole doublets assigned to $FeN_x$ sites and Fe–N(O) interaction in $FeN_x$ sites, respectively, without any detectable spectroscopic signal from Fe clusters.

NORR electrocatalysis on the FeNC-dry-0.5 was measured in a NO-saturated 0.1 M $HClO_4$ electrolyte. A linear sweep voltammetry (LSV) identifies two reduction waves (henceforth referred to as the 1st and 2nd reduction regions) before reaching an apparent diffusion-limited current density ($j_d$) of ca. 5.2 mA cm$^{-2}$ (Fig. 1a and Supplementary Fig. 6). The profile of the polarisation curve seems to indicate that the reaction follows at least two different pathways depending on the applied potential, resulting in different products. NORR can result in four products, namely $NH_3$ ($NH_4^+$ in acid), $NH_2OH$ ($NH_3OH^+$ in acid), $N_2$, and $N_2O$ ($E^0 = 0.73$, 0.38, 1.68, and 1.59 V vs. reversible hydrogen electrode (RHE), respectively). In order to identify the gaseous products formed during NORR, online differential electrochemical mass spectrometry (DEMS) coupled with a scanning flow cell (SFC; Supplementary Fig. 7) was introduced. The result showed that NO dissolved in the electrolyte started being consumed at a potential below 0.6 $V_{RHE}$ (Fig. 1b), in line with the onset potential of NORR observed on the voltammetry (Fig. 1a). Concurrently, increasing $N_2O$ formation with decreasing potential was monitored, reaching a maximum at 0.3 $V_{RHE}$ (corresponding to the 1st reduction region in voltammetry) but thereafter decreasing for further decreasing potential, becoming hardly detectable below 0.1 $V_{RHE}$. $N_2$ evolution was not discernable, while tiny level of $H_2$ byproduct could be seen at $-0.2$ $V_{RHE}$. $N_2O$ formation at 0.5–0.15 $V_{RHE}$ was also observed by SFC/DEMS study with a potentiodynamic protocol (Supplementary Fig. 8). NORR in the 1st reduction region on FeNC-dry-0.5 was compared with that on polycrystalline Pt, known to selectively form $N_2O$ at low overpotential region[11,25,37]. Pt exhibits an onset potential of ca. 0.7 $V_{RHE}$ and a $j_d$-value of ca. 1.8 mA cm$^{-2}$ (Fig. 1a and Supplementary Fig. 6), corresponding to one-electron reduction of NO to $N_2O$, independently confirmed by SFC/DEMS (Supplementary Fig. 9). Hence, similar values for the current density at ca. 0.2 $V_{RHE}$ on FeNC-dry-0.5 and for the well-defined $j_d$ observed on Pt, combined with the non-existence of other one-electron reduction products of NO, indicate that NO-to-$N_2O$

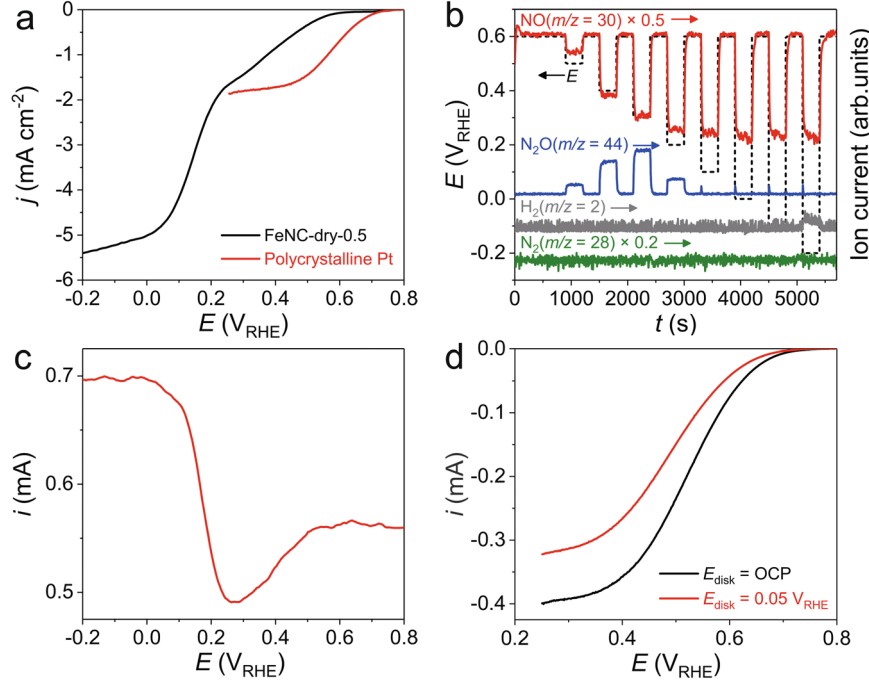

**Fig. 1 NORR electrocatalysis on FeNC-dry-0.5 studied in a half-cell system. a** NORR polarisation curves of FeNC-dry-0.5 and polycrystalline Pt electrodes. **b** Online SFC/DEMS result of FeNC-dry-0.5 during the NORR at chronoamperometry polarisations between 0.6 and $-0.2$ $V_{RHE}$. **c, d** NORR selectivity analyses using a RRDE. Pt ring current at a constant potential of 1.4 $V_{RHE}$ with a cathodic scan of FeNC-dry-0.5 disk electrode from 0.8 to $-0.2$ $V_{RHE}$ (**c**). LSV curves of Pt ring electrode from 0.8 to 0.25 $V_{RHE}$ with a constant potential of 0.05 $V_{RHE}$ and open circuit potential (OCP; ca. 0.8 $V_{RHE}$) applied on FeNC-dry-0.5 disk electrode (**d**). All the RDE and RRDE studies were performed at a 1600 rpm rotation speed in a NO-saturated 0.1 M HClO$_4$ electrolyte.

conversion is the main NORR pathway on FeNC-dry-0.5 in the 1st reduction region.

Below 0.2 $V_{RHE}$, however, the NORR current density ($j$) further increases on FeNC-dry-0.5 and the polarisation curve shows a well-defined plateau at <0.05 $V_{RHE}$ (Fig. 1a). The $j_d$ value is ca. 5.2 ($\pm$0.2) mA cm$^{-2}$, which is approximately three times the $j_d$ value observed for Pt. This suggests that the main NORR product in this region is a three-electron reduction of NO (Supplementary Note 2), corresponding to NH$_2$OH. To confirm this, rotating ring disk electrode (RRDE) experiments were carried out. The Pt ring oxidises both NH$_2$OH and NO at >0.9 $V_{RHE}$ (Supplementary Fig. 10), while NH$_3$ oxidation is almost inactive, in agreement with previous studies[21,22,38]. During LSV of the FeNC-dry-0.5 disk in the NO-saturated electrolyte, the Pt ring current (i) was recorded at a constant potential of 1.4 $V_{RHE}$ (Fig. 1c). The plateau of Pt ring current of ca. 0.55 mA at disk potentials >0.6 $V_{RHE}$ (no NORR on FeNC-dry-0.5), is attributed to NO oxidation on the Pt ring. The ring current decreases when the disk potential is polarised within the 1st reduction region, due to NO consumption on the disk (i.e., NO-to-N$_2$O conversion). When the disk potential is set within the 2nd reduction region, however, the Pt ring current increases again, reaching an absolute value even surpassing that observed when the NO concentration in the electrolyte is maximum (i.e., no NORR, >0.6 $V_{RHE}$). This indicates that the NORR products formed on FeNC-dry-0.5 in the 2nd reduction region are oxidisable on the Pt ring, which according to the potential window should be associated with NH$_2$OH rather than H$_2$ or N$_2$O. Here, NO-to-N$_2$H$_4$ conversion and its subsequent oxidation on the Pt ring could also be ruled out because of no considerable Pt ring current during the RRDE study performed at a Pt ring potential of 0.8 $V_{RHE}$, at which only N$_2$H$_4$ (not NO, NH$_3$, and NH$_2$OH) can be oxidised (Supplementary Fig. 10).

However, these analytical approaches failed to provide quantitative information in NORR selectivity because of the unknown number of electrons transferred during NH$_2$OH oxidation on Pt. Thus, we compared LSV responses of the Pt ring with and without concurrent NORR on the FeNC-dry-0.5 disk (Fig. 1d)[21]. On the Pt ring, NO is reduced to N$_2$O in 0.8–0.25 $V_{RHE}$[25], but reductions of other species (i.e., NH$_2$OH and NH$_3$) are inactive (Supplementary Fig. 10). When the disk is polarised at 0.05 $V_{RHE}$ and consumes the NO, NO reduction current on the Pt ring decreases by ca. 0.08 mA due to consequent decrease in local concentration of NO at the ring electrode. Assuming 100% NH$_2$OH selectivity on the FeNC-dry-0.5 disk, the current decrement corresponds to a collection efficiency of ca. 0.42 (Supplementary Note 3), which is in agreement with the value we calibrated (Supplementary Fig. 10). Therefore, a series of SFC/DEMS and R(R)DE studies confirms that N$_2$O and NH$_2$OH are main products on the FeNC-dry-0.5 at the 1st and 2nd NORR regions, respectively.

**Confirmation of the nature of the active site**. To understand the nature of catalytic sites in NORR, we introduced a set of Fe–N–C catalysts comprising different contents of FeN$_x$C$_y$ moieties and bulk Fe particles. The control catalysts were named 'FeNC-dry-1' and 'FeNC-wet-1', which were prepared as FeNC-dry-0.5 but with a two-fold higher Fe content in the precursor mixture and, for FeNC-wet-1, addition of a step for the aqueous complexation of Fe and phen, before milling the dried catalyst precursor (see details in Methods section)[35]. A distinct property of the control catalysts compared to FeNC-dry-0.5 is the presence of metallic iron and Fe$_3$C (Supplementary Note 1). The quantitative analysis of their $^{57}$Fe Mössbauer spectra identified that FeNC-dry-1 contains only ca. 0.2 wt% Fe particles and 2.8 wt% FeN$_x$C$_y$

moieties while FeNC-wet-1 contains *ca* 1.2 wt% Fe particles and 2.2 wt% $FeN_xC_y$ moieties (Supplementary Table 1). Due to the ability of Fe particles to catalyse graphitisation at the pyrolysis temperature, such Fe particles are surrounded by graphene shells (Supplementary Fig. 1), partially protecting them from immediate dissolution in acid medium. A N-doped carbon without any Fe intentionally added during synthesis (named 'NC') was also investigated as a third control.

LSV measurements for all the Fe–N–C catalysts revealed considerable NORR activity, while the NC produced a significantly lower current which did not reach the $j_d$ for $N_2O$ formation (i.e., ca. 1.8 mA cm$^{-2}$) even at $-0.2$ V$_{RHE}$ (Fig. 2a). In addition, SFC/DEMS analysis showed $N_2O$ production on NC over the whole potential range, while it was limited to the 1st reduction region on all the Fe–N–C catalysts (Fig. 2b and Supplementary Fig. 8). Overall, this reveals that Fe plays a pivotal role in critically enhancing the NORR activity, enabling also the formation of highly reduced products such as $NH_2OH$. Otherwise, comparable NORR polarisation curves among the all Fe–N–C catalysts suggest an insignificant catalytic role of Fe particles. From the high NORR activity of FeNC-dry-0.5 (solely consisted with $FeN_xC_y$ moieties) and its significant deactivation in the presence of cyanide anion (Fig. 2a), the isolated Fe moieties were thus indicated as the main catalytic sites in NORR[39].

In addition, a potential-dependent shift in the Fe K-edge X-ray absorption near edge structure (XANES) spectra of FeNC-dry-0.5 was identified (Fig. 2c), similar to previous findings on other Fe–N–C catalysts in oxygen reduction reaction (ORR)[40]. Coupled with voltammetric signals of electrochemical redox transition at ca. 0.6 V$_{RHE}$ (Supplementary Fig. 12), the spectral change refers to an average modification of the oxidation state from $Fe^{III}N_xC_y$ (for surface located moieties) to $Fe^{II}N_xC_y$ under NORR conditions, evidencing that the latter is the NORR active sites, similar as for molecular Fe catalysts[7,21,22,28].

**pH-dependence and product selectivity.** Motivated from the pH-dependent NORR selectivity of heme-immobilised electrode[21–23], NORR electrocatalysis of FeNC-dry-0.5 under various pH conditions was investigated. At an electrolyte pH 0, the NORR polarisation is highly suppressed in the 1st reduction region (Fig. 2d). As the electrolyte pH increases, however, NORR in the 1st reduction region becomes magnified with a substantial activity decay in the 2nd reduction region. For a quantitative comparison, we depicted the NORR activity in each reduction region by the half-wave potentials ($E_{1/2}$; Supplementary Fig. 13), which were then plotted as a function of the electrolyte pH (Fig. 2e). On an RHE scale, a linear change of the $E_{1/2}$ value can be seen for the 1st reduction region with a slope of ca. 50 mV pH$^{-1}$, corresponding to a pH-independence of the NO-to-$N_2O$ pathway. In contrast, we found almost identical $E_{1/2}$ values in the 2nd reduction region (NO-to-$NH_2OH$ pathway) at pH 0 and 1, suggesting a Nernstian behaviour with a proton transfer (PT) to be limiting the conversion. This trend is qualitatively consistent with that on heme-based electrocatalysts[21–23], showing pH-dependent/independent NORR to $NH_2OH/N_2O$, respectively.

Interestingly, however, the pH-$E_{1/2}$ correlation in the 2nd reduction region reveals an unusual negative slope at pH > 1. Along with the LSV data showing significant decrement of $j_d$ in the 2nd reduction region as the pH increases (pH 2 and 3 in particular; Fig. 2d), the negative slope suggests that decreased proton concentration significantly slows down the NO-to-$NH_2OH$ pathway beyond the expected Nernstian behaviour. This is supported by the SFC/DEMS study, showing almost no $N_2O$ signal at pH 0 throughout the entire potential range, but significant $N_2O$ signal over a broad potential range (even at $-0.2$ V$_{RHE}$ at pH 3) as the pH increases (Fig. 2f and Supplementary Fig. 14). This indicates unfavourable $NH_2OH$ formation at high pH. Therefore, it can be concluded that $N_2O$ and $NH_2OH$ productions compete in NORR electrocatalysis by FeNC-dry-0.5,

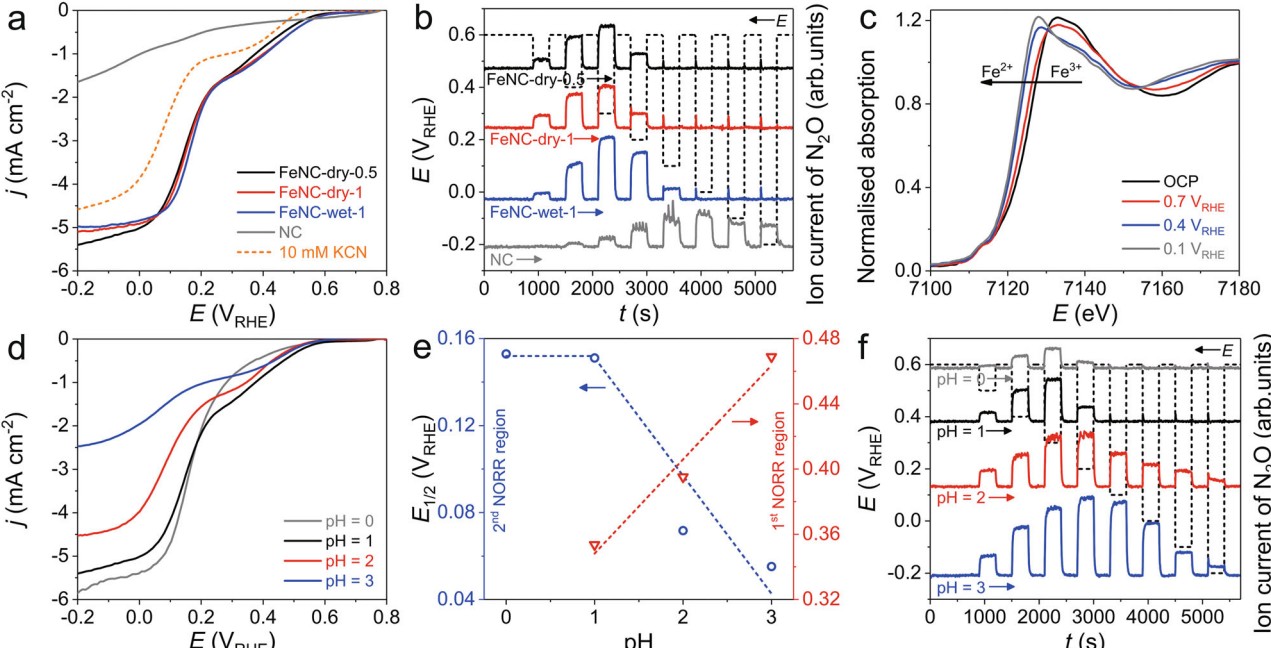

**Fig. 2 Determination of catalytic active site and pH-dependency of NORR selectivity. a** NORR polarisation curves of FeNC-dry-0.5, FeNC-dry-1, FeNC-wet-1, and NC measured in a NO-saturated 0.1 M HClO$_4$ electrolyte. Poisoning of FeNC-dry-0.5 was also examined with 10 mM KCN. **b** N$_2$O formation detected by online SFC/DEMS. **c** In situ XANES results of FeNC-dry-0.5 polarised at 0.1, 0.4, 0.7 V$_{RHE}$, and OCP (see details in Supplementary Fig. 11). **d** NORR polarisation curves of FeNC-dry-0.5 measured at various electrolyte pH 0–3. **e** Correlations of the electrolyte pH with $E_{1/2}$ estimated at the 1st and 2nd NORR regions. **f** Effects of electrolyte pH on N$_2$O formation of FeNC-dry-0.5 measured by online SFC/DEMS.

and their selectivity is strongly influenced by both the electrode potential and the electrolyte pH.

**NORR mechanism and the origin of non-Nernstian behaviour.**
To understand the NORR mechanism, we performed advanced DFT calculations. For details, the reader is referred to Supplementary Note 4. In short, we relaxed several adsorbate geometries on the Generalised Gradient Approximation level using the revised Perdew–Burke–Ernzerhof (RPBE) functional[41], where the effect of double-layer charging was included by means of an implicit solvent and a planar counter charge description for the Helmholtz layer (see Supplementary Fig. 15 for the optimised structures). This provided intermediate binding free energies as a function of surface charge density/potential (Supplementary Table 2 and Supplementary Figs. 16 and 17). Adsorption energies were further corrected by use of the hybrid Heyd–Scuseria–Ernzerhof (HSE06) functional (see Supplementary Note 4 including Supplementary Figs. 17–20 for a full sensitivity analysis of the functional choice)[42], which were then employed to develop a micro-kinetic model for the prediction of polarisation curves as a function of pH (see details in Methods and Supplementary Note 4). The calculated partial current densities for producing $NH_2OH$, $N_2O$, and $NH_3$ at pH 0 and 3 are shown in Fig. 3a as a function of applied potential on an RHE scale (see also Supplementary Note 4 for a discussion of the kinetic model sensitivity including Supplementary Figs. 21 and 22). Also, the corresponding reaction mechanism with rate-limiting steps is shown in Fig. 3b as obtained from the following analysis of the micro-kinetic modelling results and the free energy diagram in Fig. 3c (see also free energy diagram and rate-limiting step analysis in Supplementary Fig. 19).

Using the electrochemical DFT approach and Bader charge analysis, we first found all adsorbates despite $NH_3$ and $NH_2OH$ to exhibit a partially negative charge under negative electrode polarisation which we indicate by the superscript 'δ−' (see Supplementary Fig. 23 for the Bader charge analysis)[43]. In addition, the in situ XANES measurement (Fig. 2c) indicated an oxidation state of 2+ for the Fe centre. Bader charge calculations showed that the partial charge of the Fe centre marginally varies upon charging the surface (Supplementary Fig. 24). We thus conclude that Fe is always in the oxidation state 2+ throughout the reaction and denote e.g. the adsorption state of NO with $Fe^{II}–NO^{δ−}$.

From micro-kinetic modelling and a degree of rate-control analysis (Supplementary Fig. 19), we found $N_2O$ production to be limited by $Fe^{II}–N_2O_2^{δ−}$ formation. Previously, a decoupled electron transfer step to form $NO^−$ was suggested as rate-determining step (RDS) for $N_2O$ production[22]. Electron transfer on conducting materials is, however, likely too fast to resemble a RDS[44]. Instead, we suggest here a modified scenario based on charge redistribution driven by double layer electric field interaction. At the potential of zero charge (PZC), we find the NO adsorbed to the iron centre is nearly neutral judging based on the Fe–N–O angle of 150° (Supplementary Fig. 24). In going to more negative potentials, however, the additional charge yields to a partially reduced NO (Supplementary Figs. 23 and 24). In consistence with the previous understanding that $NO^−$ can easily form an N–N bond with another NO molecule[45], NO coupling becomes energetically favourable when the system and consequently also the NO is more negatively charged[46–48]. This leads to an inversion of relative energies of the $Fe^{II}–NO^{δ−}$ state and the $Fe^{II}–N_2O_2^{δ−}$ state (Supplementary Fig. 16).

To support this mechanism, we performed in situ attenuated total reflection-surface enhanced infrared absorption spectroscopy (ATR-SEIRAS) studies (Fig. 3e–g and Supplementary Fig. 25). The ATR-SEIRAS spectra identified two main bands at

ca. 1723 (the high frequency NO; $NO_{High}$) and 1685 cm$^{-1}$ (the low frequency NO; $NO_{Low}$) at −0.2 $V_{RHE}$ (Fig. 3e), both of which showed a Stark effect with the slope of ca. 50 cm$^{-1}$ V$^{-1}$. In addition, the positions of both bands were unchanged by solvent isotope labelling (in $H_2O$ and $D_2O$ solutions; Fig. 3f), indicating that these bands are associated with nonprotonated species. IR bands of organometallic Fe-porphyrin complexes observed at ca. 1700 cm$^{-1}$ have usually been assigned to the $Fe(\eta^1\text{-NO})$, where NO is bonded to Fe via nitrogen[49,50]. Thus, we assigned these to the adsorbed NO species on the Fe centre, i.e., $Fe^{II}–NO^{δ−}$. Also, considering the high sensitivity of the NO vibration frequency to the local chemical environment[51–53], the presence of two separate bands implies that there exist (at least) two chemically inequivalent $Fe^{II}–NO^{δ−}$ species. However, these bands are separated by only ca. 40 cm$^{-1}$, and no appreciable signal was shown below 1600 cm$^{-1}$. Thus, the possibility of different binding modes such as $\eta^1$-ON and $\eta^2$-NO could reasonably be excluded[54]. Instead, slightly more reduced NO forming a more bent Fe–N–O geometry may explain the band at the lower frequency ($NO_{Low}$). Furthermore, the integrated peak intensities of both bands increased with decreasing an applied bias, inferring the increase of more $Fe^{II}–NO^{δ−}$ species at lower potential (Fig. 3g). Thus, the key intermediate of $Fe^{II}–NO^{δ−}$, which is predicted to exist with a high coverage below 0.2 $V_{RHE}$ (Supplementary Fig. 19), is spectroscopically confirmed.

Since the formation of $Fe^{II}–N_2O_2^{δ−}$ does not involve a PT, it is pH-independent on a SHE scale (Supplementary Fig. 21). On an RHE scale, we thus saw the expected Nernstian overpotential shift of ca. 59 mV × ΔpH = 178 mV when going from pH 0 to 3 (Fig. 3a), in agreement with the experimental results in the 1st reduction region (Fig. 2e). The production of $NH_2OH$ and $NH_3$, however, is limited by a proton-coupled electron transfer (PCET) step to $Fe^{II}–NO^{δ−}$ (Supplementary Fig. 19), and thus showed a strong pH-dependence on an SHE scale (Supplementary Fig. 21). Interestingly, however, we found the pH-induced overpotential shift to be larger than the expected 178 mV. In consequence, even after correcting for this shift by plotting the data on an RHE scale (Fig. 3a), we still observed a decrease of the reaction rate with increasing pH.

Such a non-Nernstian behaviour is originated from differences in the surface charge dependence of intermediate binding energies (Supplementary Fig. 16). At a higher pH (for a fixed potential on RHE scale), the corresponding potential in the SHE scale is more negative, which increases the surface charge. The stronger stabilisation of $Fe^{II}–NO^{δ−}$ with negative charge relative to the following $Fe^{II}–NHO^{δ−}$ state (Supplementary Fig. 16), increases the overall reaction barrier at fixed RHE potential (Fig. 3c). This super-Nernstian decrease of the reaction rate with increasing pH explains the experimentally observed pH dependence (Fig. 2e) and underlines the role of surface charge in controlling product selectivity.

We further elucidate that the redox non-innocent property of nitrosyl ligand originates the particularly strong surface charge stabilisation of the NO* intermediate. With changing the redox state of nitrosyl ligand from $NO^+$ to NO to $NO^−$, the metal–N–O angle is known to vary from the linear geometry (as stabilised by the π-back bond; Supplementary Fig. 26) to the bent geometry (as stabilised by the σ-forward bond; Fig. 3d (RPBE) and Supplementary Fig. 27 (HSE06))[55]. When the surface is negatively charged, the Fermi energy increases and the occupation of the NO π* orbital also increases, yielding an enhanced $NO^−$ character as evidenced from the decrease in Fe–N–O angle (Supplementary Fig. 24). At higher cathodic overpotential, thus, the nitrosyl ligand becomes more reduced, and thus more effectively stabilised by forming the stronger Fe–N σ-forward bond. This results in the surface charge dependence of the

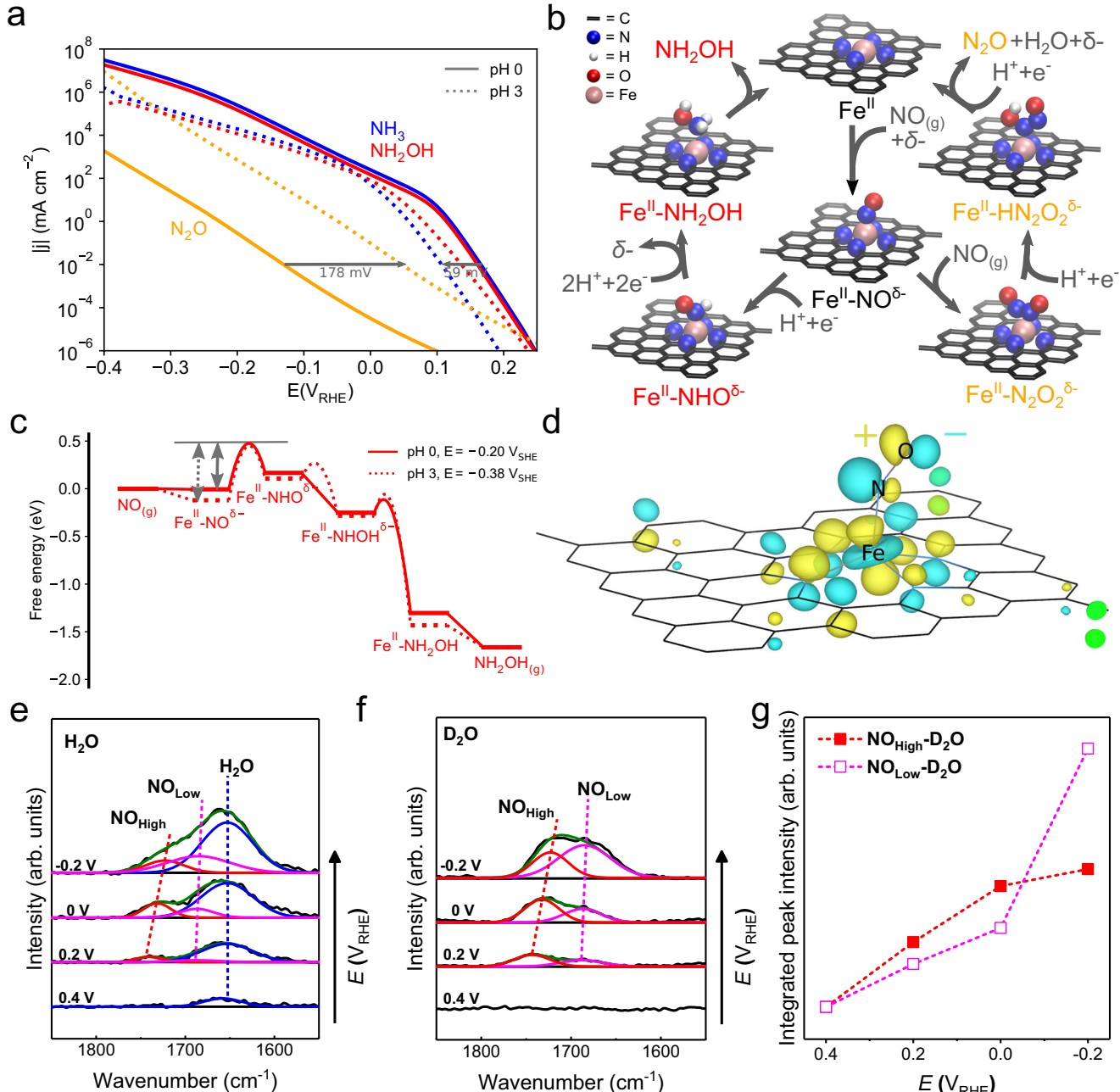

**Fig. 3 First-principles based electrochemical multiscale simulation studies and in situ ATR-SEIRAS analyses on the mechanism of NORR at graphene-embedded Fe–N$_4$ moiety. a** Polarisation curves from micro-kinetic modelling relying on HSE06-based adsorption energies showing the partial current densities of three different NORR products for pH 0 and pH 3 on an RHE scale. Horizontal arrows depict the overpotential shift with pH. **b** Reaction mechanism with the key rate-limiting intermediates as obtained from analysis of the micro-kinetic modelling results. **c** Free energy diagram for NORR to NH$_2$OH (HSE06 level) at $-0.2$ V$_{RHE}$ for two different pH values. The grey arrows indicate that the more negative corresponding potential on an SHE scale at pH 3 leads to an additional stabilisation of Fe$^{II}$–NO$^{\delta-}$ relative to the transition state towards the Fe$^{II}$–NHO$^{\delta-}$ state on an RHE scale. The x-axis corresponds to the overall reaction coordinate being decomposed into the elementary steps (containing both PCET and chemical steps). The free energy diagram has been evaluated without accounting for pressures and coverages. **d** Spin-up HOMO of the Fe$^{II}$–NO$^{\delta-}$ state at the PZC (at the $\Gamma$-point and an isovalue of $\pm1.5\times10^{-3}$) (RPBE level). In situ ATR-SEIRAS analysis of FeNC-dry-0.5 measured in NO-saturated 1 mM HClO$_4$ + 0.1 M KClO$_4$/H$_2$O (**e**) and 1 mM DClO$_4$ + 0.1 M KClO$_4$/D$_2$O (**f**) electrolytes. **g** Integrated peak intensity of NO$_{High}$ and NO$_{Low}$ measured in the 1 mM DClO$_4$ + 0.1 M KClO$_4$/D$_2$O electrolyte. The IR spectra were collected at constant potentials of 0.4, 0.2, 0, and $-0.2$ V$_{RHE}$ with a reference spectrum at 0.8 V$_{RHE}$.

Fe$^{II}$–NO$^{\delta-}$ intermediate, yielding the super-Nernstian behaviour during NH$_2$OH production.

Additionally, it is important to mention that the presence of a finite surface charge does not always stabilise the adsorbate states (Supplementary Fig. 16). As an example, unlikely to the Fe$^{II}$–NO$^{\delta-}$

state, the Fe$^{II}$–NH$_3$ state has an anti-bonding singly occupied (highest) MO (SOMO) (Supplementary Fig. 28). Filling of this SOMO results in a destabilisation at more negative potentials. This indicates that charge stabilisation is a complex function of the electronic structure of adsorption states, and thus a quantum-

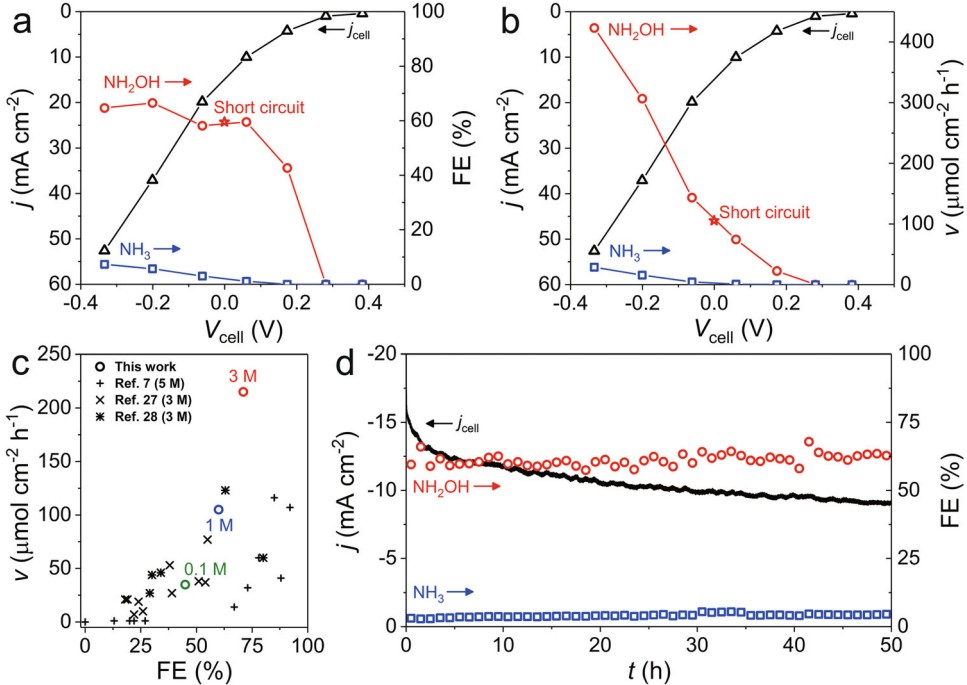

**Fig. 4 NH₂OH production in a flow-type H₂–NO fuel cell. a** FE and **b** production rate (*v*) of NH₂OH and NH₃ at various cell voltages: (conditions) an Ar-saturated 1 M HClO₄ electrolyte, Pt anode, FeNC-dry-0.5 cathode. The fuel cell operation was performed for 1 h at a constant cathode potential, screened from 0.4 to −0.2 V$_{RHE}$ with a 0.1 V potential step. **c** Comparison of NH₂OH production rate and FE$_{NH2OH}$ for various catalysts recorded in H₂–NO fuel cells at short-circuit conditions. The electrolyte used for each measurement was indicated in the figure. **d** Current density and FEs during a 50 h long-term durability test at a short-circuit condition.

mechanical level of simulation reflecting the finite bias potential effect is a necessity for a comprehensive understanding of the electrochemical reaction mechanism.

**NH₂OH production in a flow-type H₂–NO fuel cell.** To confirm the NH₂OH production in a fuel cell with FeNC-dry-0.5, a H-type flow cell with gas-diffusion electrode (GDE) was employed as a prototype reactor (Supplementary Fig. 29). An Ar-saturated 1 M HClO₄ electrolyte was introduced into the cathode compartment, directly connected to ion chromatography (IC) for online monitoring of NH₃ and NH₂OH production. NH₂OH production is detected at a cell voltage ($V_{cell}$) below 0.17 V ($E_{cathode} = 0.2$ V$_{RHE}$), and its Faradaic efficiency (FE$_{NH2OH}$) reaches 60–70% as $V_{cell}$ further decreases (Fig. 4a, b and Supplementary Fig. 30). NH₃ production is also found at $V_{cell}$ below 0.06 V ($E_{cathode} = 0.1$ V$_{RHE}$), while FE$_{NH3}$ is as low as <7%, verifying highly selective NO-to-NH₂OH conversion on FeNC-dry-0.5 over NO-to-NH₃.

However, the overall FE of NH₂OH and NH₃ is lower than 75%. The remainder is consequently assigned to N₂O production, which becomes magnified or diminished as the electrolyte acidity decreases (0.1 M; Supplementary Fig. 31) or increases (3 M; Supplementary Fig. 32), respectively. This result is qualitatively in accordance with the data measured in the half-cell setup (Fig. 2d–f). Despite the enhanced N₂O formation, FeNC-dry-0.5 demonstrates a successful NH₂OH production of ca. 105 μmol h⁻¹ cm⁻² at a short-circuit condition (i.e., $V_{cell} = 0$ V), which is one of the highest values reported in literature (Fig. 4c and Supplementary Table 3), although much milder condition (1 M HClO₄) was employed in this work. Notably, the cell operation with a strong acid electrolyte of 3 M HClO₄, i.e., similar condition with that in literature, verifies remarkable NH₂OH production rates of ca. 215 and 519 μmol h⁻¹ cm⁻² at short-circuit and electrolysis ($V_{cell} = -0.24$ V) modes, respectively.

A durability test at the short-circuit condition reveals an initial current density of ca. −16 mA cm⁻² (Fig. 4d), while it rapidly declines by ca. 25% for the first 5 h operation and the decay is mitigated afterward. However, NORR selectivity is almost untouched throughout the 50 h period, showing a stable FE$_{NH2OH}$ of ca. 61%. Considering a fact that a destruction of active FeN$_x$C$_y$ moieties or nearby carbon surface significantly alters reaction selectivity as well-exemplified in ORR and CO₂ reduction cases[56,57], catalytic degradation could be ruled out as a major cause of the current density decay. The good catalytic stability of FeNC-dry-0.5 was also corroborated by the RDE studies (Supplementary Fig. 33), showing stable NORR over 10 h measurement. Meanwhile, a fully wetted catalyst layer (i.e., decrement in its hydrophobicity) and small leakage of electrolyte through the GDE were found after the durability test (Supplementary Fig. 34). This suggests that the initial current density decay may be attributed to partial electrolyte flooding into the GDE, leading to the blockage of diffusion path for NO gas and consequently to the partial loss of the triple-phase-boundary. However, the single-cell operation with this prototype reactor successfully validates the potential of practical NO-to-NH₂OH conversion on the FeN$_x$C$_y$ moieties with high selectivity and catalytic stability, and brings hope for durable operation in a device-level if rational systematic strategies minimising the electrolyte flooding in GDE are developed[58,59].

**Discussion**

In summary, we presented a novel single-atom Fe catalyst for efficient NH₂OH production from electrochemical NO reduction. By performing detailed electrochemical analysis, we identified the catalytic site to be electrogenerated Fe$^{II}$N$_x$C$_y$ moieties. From a combination of in operando spectroscopy and electrochemical hybrid-level DFT-based multiscale modelling, we further obtained full mechanistic details about the NH₂OH and N₂O

production pathways and their pH dependence. Further, we revealed an intriguing super-Nernstian pH dependence of the $NH_2OH$ pathway which originates from the redox non-innocent character of NO. The resulting surface charge sensitivity of the $Fe^{II}-NO^{\delta-}$ state leads to an increased sensitivity to the over-potential changes that occur by varying the pH conditions. We then finally validated the potential of practical NO-to-$NH_2OH$ conversion on the isolated $FeN_xC_y$ moieties also in a fuel cell device showing unprecedented long-term stability and performance. Along with the obtained detailed mechanistic insights, which will be invaluable for the development of future NO reduction catalysts, the presented excellent performance metrics in device-level will provide an important stepping stone towards the technological development of fully sustainable electro-synthesis of valued nitrogen products from electrochemical nitrate/nitrite denitrification, which is environmentally vital for balancing the disturbed global nitrogen-cycle.

## Methods

**Catalyst synthesis.** FeNC-dry-0.5 and FeNC-dry-1 catalysts were prepared from the Fe$^{II}$ acetate (95%, Sigma-Aldrich), phen (≥99%, Sigma-Aldrich), and ZIF-8 ($ZnN_4C_8H_{12}$, Basolite Z1200 from BASF)[36]. The precursor mixture (1 g), containing 0.5 and 1.0 wt% Fe with a mass ratio phen/ZIF-8 of 20/80, was homogenised by dry ball-milling. The milling was conducted in a $ZrO_2$ crucible with 100 $ZrO_2$ balls (5 mm diameter) using a planetary ball-miller (FRITSCH Pulverisette 7 Premium) for four cycles of 30 min at 400 rpm. The resulting catalyst precursor was pyrolysed at 1323 K in Ar (5N, Daedeok) for 1 h, leading to FeNC-dry-0.5 and FeNC-dry-1. FeNC-wet-1 catalyst was prepared identically to the FeNC-dry-1 except for the addition of wet-impregnation step before the homogenised process[35]. Catalyst precursors were dissolved in a mixture of ethanol/water solution (1/2 vol. ratio) to form Fe(phen)$_3$ complex. Precursor powder gathered by the solvent evaporation was then ball-milled and pyrolysed, yielding FeNC-wet-1 after acid-washing with pH 1 $H_2SO_4$. The Fe content in the catalysts after pyrolysis was measured by inductively coupled plasma mass spectrometry (NexION I, Perkin-nElmer), and found to be ca. 1.5, 3.0, and 3.4 wt% for FeNC-dry-0.5, FeNC-dry-1, FeNC-wet-1, respectively. Active site density of FeNC-dry-0.5 estimated with the in situ nitrite poisoning method[60]. For the synthesis of NC catalyst, free of Fe species, the precursor powder composed of phen and ZIF-8 (without Fe$^{II}$ acetate), was dry ball-milled and pyrolysed as methods for the dry Fe–N–C catalysts. Due to trace amount of Fe impurity in the commercial ZIF-8 (>100 ppm)[61], Fe-free ZIF-8, synthesised by mixing 2-methylimidazole (2-MeIm) and Zn nitrate hexahydrate (Zn salt) in aqueous solution (molar ratio, Zn salt : 2-MeIm:water = 1:60:2228)[62], was used for the NC preparation.

**Catalyst characterisation.** X-ray diffraction (XRD) patterns were collected with a high resolution X-ray diffractometer (X'Pert PRO MPD, PANalytical) equipped with a Cu Kα X-ray source. The XRD measurements were performed at an accelerating voltage of 45 kV and a current of 55 mA with a scan rate of 2° min$^{-1}$. Raman spectra were measured by a NRS-5100 (JASCO) with a 532 nm laser excitation. Transmission electron microscopy (TEM) and energy-dispersive X-ray spectroscopy analyses were carried out using a JEM-2100F (JEOL LTD.) at 200 kV. The diluted aqueous solution of Fe–N–C catalysts was deposited on a Ni mesh grid coated with a carbon film (CF150-Ni, Electron Microscopy Science). X-ray photoelectron spectroscopy signals were collected with a Sigma Probe (Thermo VG Scientific) equipped with a microfocused monochromator X-ray source. $^{57}$Fe Mössbauer spectra were obtained at room temperature with a $^{57}$Co source in rhodium. The spectrometer was operated with a triangular velocity waveform, and a NaI scintillation detector was employed for the γ-ray detection. Calibration was performed with an α-Fe foil. The Fourier transforms of the EXAFS signals were analysed from Fe K-edge X-ray absorption spectra collected at room temperature at the SAMBA beamline (Synchrotron SOLEIL).

**Electrochemical characterisation.** The electrochemical measurements were conducted with a VMP3 potentiostat (Bio-Logic) in a three-electrode glass cell with an electrode rotator (RRDE-3A, ALS). Graphite rod and saturated Ag/AgCl (RE-1A, EC-Frontier) electrodes were used as counter and reference electrodes, respectively. The catalyst ink was prepared by dispersing 5 mg catalyst in Nafion solution (2713 μL water, 221 μL isopropyl alcohol (IPA), and 50 μL Nafion solution (5 wt%)). The homogenised catalyst ink (15 μL) was drop-casted on glassy carbon (GC, 0.126 cm²) of working electrode (012613, ALS). The ink on the electrode was dried at room temperature and the resulting loading amount was ca. 200 μg cm$^{-2}$. For a comparison, a commercial polycrystalline Pt electrode (0.07 cm², 011170, ALS) was employed. Before the electrochemical experiments, the reference electrode was calibrated against a Pt electrode in a $H_2$-saturated (5N, Daedeok) electrolyte to correctly convert potentials to the RHE scale. Reference electrode was

doubly separated with a glass bridge tube to avoid the halogen contamination. In general, 0.1 M $HClO_4$ (pH 1) solution was used as an electrolyte, which was prepared from ultrapure water (>18.2 MΩ, Arium® mini, Sartorius) and concentrated $HClO_4$ (70%, Sigma-Aldrich). For the studies about the pH effects, 1 M (pH 0), 0.01 M (pH 2), and 0.001 M (pH 3) $HClO_4$ electrolytes were employed, after adjusting the ionic strength to 0.1 M with a $KClO_4$ salt (≥99%, Sigma-Aldrich) for pH 2 and 3 due to their poor ionic conductivity. After deaeration by Ar-bubbling, the electrolyte was saturated with NO gas (3N, Daedeok), the possible $NO_2$ impurity of which was removed by two glass bubblers filled with a 4 M KOH (85%, Daejung) solution[21]. A headspace of the electrochemical cell was protected by Ar flow to prevent undesirable $NO_2$ formation and $O_2$ dissolution.

NORR polarisation was investigated with a 10 mV s$^{-1}$ scan rate at a 1600 rpm rotation speed in a potential range from −0.2 to 0.8 V$_{RHE}$. The polarisation curves were shown after subtractions of the capacitive currents, which were measured in Ar-saturated electrolytes. For the poisoning test, NO-saturated 0.1 M $HClO_4$ electrolyte dissolving 10 mM KCN (≥96%, Sigma-Aldrich) was used and the electrolyte was handled very carefully. Prior to the RRDE experiments, cyclic voltammograms (CVs) of the Pt ring electrode (10 mV s$^{-1}$, 1600 rpm) were gathered in Ar-saturated 0.1 M $HClO_4$ dissolving either 1.5 mM $NH_3$ (25 wt%, Merck), 1.5 mM $NH_2OH$ (50 wt%, Sigma-Aldrich), or 1.5 mM $N_2H_4$ (50 wt%, Sigma-Aldrich) and in NO-saturated 0.1 M $HClO_4$. A current from the Pt ring, which was polarised at a constant potential of 1.4 V$_{RHE}$ (or 0.8 V$_{RHE}$), was collected during a cathodic scan of FeNC-dry-0.5 disk electrode from 0.8 to −0.2 V$_{RHE}$ in NO-saturated 0.1 M $HClO_4$. Otherwise, LSV polarisation of the Pt ring from 0.8 to 0.25 V$_{RHE}$ was gathered with potential holds of the disk electrode at 0.05 V$_{RHE}$ or OCP[21]. A collection efficiency of the RRDE electrode was estimated with 2 mM $K_3[Fe(CN)_6]$ (≥99%, Sigma-Aldrich) dissolved in an Ar-saturated 0.1 M $KNO_3$ (≥99%, Sigma-Aldrich) electrolyte at rotation speeds of 100, 400, 900, 1600, and 2500 rpm. CV of the disk was measured in a potential range of 0.6 to −0.2 V$_{Ag/AgCl}$ with a potential hold of the Pt ring at 0.6 V$_{Ag/AgCl}$. The square-wave voltammetry was measured in a potential range of 0.05–1.2 V$_{RHE}$ with a step potential of 10 mV, a potential amplitude of 1 mV, and a scan frequency of 5 Hz in an Ar-saturated 0.1 M $HClO_4$ electrolyte. Chronopotentiometry of the FeNC-dry-0.5 was gathered at fixed current density of −3.5 mA cm$^{-2}$ for 10 h. For stability comparison with typical molecular catalysts, heme (>95%, TCI) and iron phthalocyanine (FePc, 90%, Sigma-Aldrich) was grafted on multi-walled carbon nanotube (Carbon Nanomaterial Technology Co.) with a 1.5 wt% Fe loading (i.e., identical to the Fe content of the FeNC-dry-0.5), and their NORR polarisation was measured. The PZC of FeNC-dry-0.5 was measured using staircase potentiostatic electrochemical impedance spectroscopy. The measurement was performed in an Ar-saturated 10 mM NaF electrolyte from −1.2 to 0.8 V$_{Ag/AgCl}$ at 10 mHz frequency and with a 10 mV potential amplitude.

**In situ and operando spectroscopic analyses.** The online DEMS studies were carried out with the SFC directly connected to mass spectroscopy (Max 300 LG, Extrel). The SFC had an U-shaped channel with an opening diameter of 1 cm at the bottom of the cell, where electrochemical contact was made with the working electrode. At the top of the cell, gas/volatile products evaporated through a hydrophobic polytetrafluoroethylene (PTFE) membrane, which was positioned ca. 100 μm away from the electrode, and introduced into the vacuum system of the mass spectrometer. Working electrode was prepared by dropping the catalyst inks onto GC electrode (0.07 cm², 011169, ALS) with a catalyst loading of 200 μg cm$^{-2}$. NO-saturated electrolytes were flowed at 0.07 mL min$^{-1}$. An Ag/AgCl reference and a graphite tube counter electrodes were connected to the SFC inlet and outlet, respectively. The DEMS studies were performed with two different potential protocols: a stepwise chronoamperometry from 0.6 to −0.2 V$_{RHE}$ and a CV at a 1 mV s$^{-1}$ scan rate in a potential range of −0.2 to 0.8 V$_{RHE}$. For a comparison, a commercial polycrystalline Pt electrode (0.07 cm²) was employed. During the measurements, ion currents from NO, $N_2O$, $H_2$, and $N_2$ were monitored at $m/z$ = 30, 44, 2, and 28, respectively. The NO signal was shown after correction of the initial signal at $m/z$ = 30 by subtraction of 27% signal from $m/z$ = 44 to remove $N_2O$ contributions at $m/z$ = 30.

The in situ XANES measurements were performed at KIST-PAL beamline (1D) at the Pohang Accelerator Laboratory (PAL). A flow-type in situ X-ray absorption spectroscopy (XAS) cell was equipped with an electrolyte flow channel and a window for X-ray radiation. The window was a carbon-coated Kapton film (200RS100, DuPont, $t$ = 0.05 mm, $A$ = 0.385 cm²), which was directly used as a working electrode after a loading of the FeNC-dry-0.5 (3 mg cm$^{-2}$). Pt wire counter and Ag/AgCl reference electrodes were connected at the electrolyte outlet. Due to safety issues in PAL, Ar-saturated 0.1 M $HClO_4$ + 1.5 mM $KNO_2$ (≥96%, Sigma-Aldrich) solution was used as an electrolyte (no direct NO-bubbling), in which nitrite can be chemically decomposed to NO at such highly acidic conditions[9,60]. The XAS spectra were collected at a fluorescence mode after the beam calibration with a Fe foil. During the XAS measurements, the FeNC-dry-0.5 was polarised at constant potentials of 0.1, 0.4, 0.7 V$_{RHE}$, and OCP with a SP-150 portable potentiostat (Bio-Logic).

The in situ ATR-SEIRAS measurements were carried out with an Au thin film-coated Si prism working electrode (Veemax, 2 cm in diameter), which was placed in a two-compartment, three-electrode spectro-electrochemical cell. The Au thin film was prepared by an electroless plating procedure[63]. The working electrode, on which FeNC-dry-0.5 catalyst was deposited, and the Ag/AgCl (Basi, 3 M NaCl)

reference electrode were separated from the Pt wire counter electrode using a Nafion 117 membrane. The cell was integrated into a Fourier transform infrared spectrophotometer (FT-IR, VERTEX 80v, Bruker) equipped with a mercury cadmium telluride detector and a variable angle specular reflectance accessory (VeemaxIII, Pike Technologies). All spectroscopic measurements were conducted at a 4 cm$^{-1}$ spectral resolution, and the spectra were presented in absorbance mode. NO-saturated 1 mM HClO$_4$ + 0.1 M KClO$_4$/H$_2$O and 1 mM DClO$_4$ + 0.1 M KClO$_4$/D$_2$O solutions were used as electrolytes. Potential-dependent IR measurements were carried out during chronoamperometry polarisations between 0.8 and $-0.2$ V$_{RHE}$. The spectrum collected at 0.8 V$_{RHE}$ was used as the baseline.

**H$_2$–NO single-cell operations**. The H-type flow cell was operated with Ar-saturated 0.1, 1, or 3 M HClO$_4$ solution as both an anolyte and a catholyte, which were separated by a Nafion 115 membrane (1.5 × 1.5 cm$^2$, DuPont). Flow rate of the electrolyte was ca. 7.3 ± 0.2 μL s$^{-1}$ for each compartment (inner volume = ca. 1.9 cm$^3$ each). On a carbon paper with a 20 wt% PTFE content (3 × 3 cm$^2$, TGP-H-090, Toray), highly hydrophobic carbon mesoporous layer (MPL) was fabricated by spraying an ink emulsion—100 mg Ketjen black EC-300J, 100 mg PTFE (60 wt%, Sigma-Aldrich), and 20 mL IPA (99.5%, Sigma-Aldrich)—and by subsequent heat-treatments at 513 and 613 K under N$_2$ atmosphere for 30 min each. The resulting MPL had a Ketjen black EC-300J loading of 2 mg cm$^{-2}$. Anode and cathode catalysts were Pt/C (37.7 wt%, TEC10V40E, TANAKA) and FeNC-dry-0.5. Catalyst inks—4 mg catalyst + 200 μL Nafion solution (5 wt%) + 2800 μL IPA—were sprayed onto the MPL to reach target catalyst loadings of 1 mg$_{Pt}$ cm$^{-2}$ and 0.7 mg cm$^{-2}$ for anode and cathode, respectively. Active catalyst area on the GDE was 1 × 1 cm$^2$, which faced to the electrolyte flow compartment. H$_2$ and 10% NO/Ar (Daedeok) gases were introduced behind the anode and cathode GDEs at a 60 sccm flow rate, which was controlled by the mass flow controllers (Line Tech).

The single-cell operation was performed at room temperature for 1 h at a constant cathode potential, screened from 0.4 to $-0.2$ V$_{RHE}$ with a 0.1 V potential step. An Ag/AgCl reference electrode was introduced in the anode electrolyte compartment to construct three-electrode cell. The cell voltage was estimated from the difference between cathode and anode potentials. For the 50 h long-term durability test, the cell was operated at a short-circuit condition ($V_{cell}$ = 0 V). Before and after the durability test, contact angle on the cathode electrode was measured by PHOENIX-300 TOUCH (SEO) contact angle analyser. To quantify NORR products, the catholyte outlet was directly connected to online IC (ICS-2100, Thermo scientific), which collected and analysed samples every 20 min. An IC Y-521 (Shodex) cation column was employed with a 4 mM nitric acid (65%, Merck) eluent. Before the single-cell operations, retention time and concentration of NH$_3$OH$^+$ and NH$_4^+$ were calibrated with standard HClO$_4$ solutions dissolving 0–0.5 mM NH$_2$OH/NH$_3$ mixtures. The IC data were analysed by using the Chromeleon 6.8 program.

**Computational methods**. DFT calculations of reaction energetics were carried out with a periodic plane-wave implementation and ultra-soft pseudo-potentials using the QUANTUM ESPRESSO version 6.1 on a single Fe–N$_4$ moiety embedded into a graphene unit cell[64]. The self-consistent continuum solvation implicit solvation model as implemented in the Environ QUANTUM ESPRESSO module was used to model the presence of implicit water[65]. The surface charge density was modulated by changing the total charge of the system and a planar counter charge was introduced above the slab to compensate the charge[48,65,66]. The relaxed surface states were also re-calculated using the Vienna Abinito Simulation Package as spin-polarised single point calculations using the RPBE+U and HSE06 functionals[42]. Assuming a constant double layer capacitance $C_{dl}$ (ca. 20 μF cm$^{-2}$ for graphene)[67,68], the surface charge density $\sigma$ is generated according to $\sigma = C_{dl}(E - E^{PZC})$, where $E$ is the applied electrode potential and $E^{PZC}$ is the PZC which we measured here to be ca. 0 V$_{SHE}$ (Supplementary Fig. 35). PCET steps were described based on the computational hydrogen electrode and electrochemical barriers were estimated from the reversible potentials of each elementary reaction step[69,70]. For more detailed information about the calculations, the reader is referred to the Supplementary Note 4.

## Data availability

The data that support the findings of this study are available from the corresponding authors upon reasonable request.

## Code availability

The QUANTUM ESPRESSO DFT program package is available from the website https://www.quantum-espresso.org/, the CatMAP micro-kinetic modeling program package from https://github.com/SUNCAT-Center/catmap. Input and output files for both simulation techniques are available from the corresponding authors upon reasonable request.

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

## Acknowledgements

This work was supported by the National Research Foundation of Korea (NRF) grant funded by the Korea government(MSIT) (No. NRF-2017M3D1A1039378, NRF-2019M3D1A1079309, and NRF-2020R1A2C4002233) and by the KIST Institutional Program.

## Author contributions

C.H.C. and H.K. (Hyungjun Kim) conceived and directed the project. D.H.K. and S.R. conducted most of experimental and computational analyses, respectively. S.K. contributed to part of the computational calculations. W.K. and B.K. contributed to the in situ ATR-SEIRAS measurements. H.K. (Haesol Kim) contributed to part of the DEMS measurements. G.B. contributed to part of catalyst synthesis. H.S.O. contributed to part of the XAS measurements. F.J. contributed to the catalyst synthesis and material characterisation. C.H.C., H.K. (Hyungjun Kim), D.H.K. and S.R. wrote the manuscript with contribution from all authors.

## Competing interests

The authors declare no competing interests.
