## [Peer Review File · Nature Communications]

Reviewer #1 (Remarks to the Author):

Summary:

In "Selective electrochemical reduction of nitric oxide to hydroxylamine by atomically dispersed iron catalyst", Kim et al. report a new single-atom Fe catalyst which is supported on graphene and stabilized with N atoms around the active site, mimicking the heme structure and is capable of reducing NO to H₂NOH with good selectivity alongside production of NH₃. The DFT calculations by SR and HK are based on an implicit solvation model coupled with the computational hydrogen electrode to construct reaction free energies diagrams and polarization curves using microkinetic analyses. It was found that the *N₂O₂ intermediate is rate-determining, and that the origin of super-Nernstian shift can be attributed to a stronger stabilization of the *NO intermediate at a more negative absolute potential. The high-quality calculations complement a thorough set of experimental analysis/characterization data and in my opinion warrants publication of this manuscript in Nature Communications after the following comments have been addressed.

Detailed comments to be addressed prior to publication:

Main Text

Figure 3a: It would be helpful to overlay experimental polarization data for H₂NOH and NH₃ alongside the calculated curves, perhaps in a separate panel so that data are not overwhelming. The scale may be different however separate y axes can be used.

Figure 3d: It is somewhat difficult to distinguish the orbital character around the *NO adsorbate. Can the authors provide a cartoon/schematic for what the orbital should look like relative to the graphene support, or otherwise outline more clearly the most relevant isosurface around *NO?

Supplementary Information

This is very interesting and warrants further elaboration. Is the choice of U = 2 eV based on adsorption energies rather than the linear response method? Can the authors provide data on their tests with hybrid functionals, and how the different intermediates are affected by the application of U, i.e. how do these energies look with RPBE alone, RPBE+U, and hybrid?

"(8) NH₃* + H+(aq) + e⁻ ⇌ [*-NH₃]‡ ⇌ NH₃(g) (chemical)"

The proton-electron pair should not appear on the left side of this equation.

*N₂O₂ is actually optimized at a surface charge of -0.4 e or more negative as the adsorbate cannot be stabilized otherwise (page S6), so this caption appears to be inaccurate.

In Supplementary Figure 16, it is seen that most of the intermediates along the NO reduction pathway respond in a similar way to surface charge, however, *NH₃, *NHO, and *NHOH are clear outliers. Can the authors provide a physical explanation for why these adsorbates have different energy dependences on applied surface charge?

Page S40, spelling error in equation “chrage”

The computed polarization curves at pH 3 and pH 0, free energy diagram at -0.20 V vs. RHE, degree of rate-control analyses, HOMO-1 plot, and optimization of the NO^* intermediate as a function of surface charge are fine to publish in their present state.

Reviewer #2 (Remarks to the Author):

The report by Kim et al concerns the use of single-site FeN_4 catalysts on graphene to investigate NO reduction reaction (NORR) pathways to N_2O and NH_2OH . The work is well done, extensive electrochemical analyses are supported by kinetic and DFT modeling; the catalytic materials were extensively characterized in several previous reports, here only characterized by XANES to predict Fe oxidation states by applied potential (other characterizations are given in supplemental but not discussed: XPS, Mossbauer, Raman, EXAFS, TEM). The results are presented in regard to the potential use of H_2 -NO fuel cell reactor, which is fairly unique but not widely considered as viable.

One issue is that this is the latest in a series of reports on the use of these FeN_4 catalysts (not entirely clean, but well characterized) to reduce H_2O_2 (Angewandte 2017) and O_2 (Energy and Environmental Sci 2018). Thus, the description of the catalyst formation and characterization in the text is limited, forcing new readers to do substantial background reading to understand the difference in materials used, e.g. FeNC-dry-1 vs FeNC-wet-1 . Indeed, the report would be much enhanced by inclusion of Likewise, the electrochemical and product analyses are based on extensive previous work, e.g. series of papers by de Groot et al using rotating and ring-disc electrodes, which themselves followed numerous other investigations of the NORR pathways using porphyrins, phthalocyanines and heme enzymes/proteins. Thus the novelty and wide impact of the work may not be to the high standard of your journal.

What is somewhat unique in this report, beyond the apparent selectivity for forming NH_2OH , is the electrochemical multiscale simulation studies and the resulting interpretation of the pathway intermediates. The authors use NO^* to designate the Fe-bound intermediates, which is distracting and should be changed. The schema in Fig 3 suggests that NO binds to the FeN_4 to generate NO^* , but does not stipulate the Fe oxidation state! Both Fe(II) and Fe(III) states of FeN_4 moieties are known to bind NO, but with differing affinities and electronic structures. Likewise, structural and electronic variations between isoelectronic Fe(I)-(NO) and Fe(II)-(NO-) are known in different systems.

From my understanding of the voltammetry and simulations, the NO^* state corresponds to the Fe(II)-NO adduct which then can undergo two different paths, a $2 e^-/\text{H}^+$ reduction to an Fe(II)-NHOH species or reaction with a second NO to form an $\text{Fe(II)-N}_2\text{O}_2$ moiety. Both of these species are uncharacterized in experimental chemistry; a model Fe(II)-NHOMe species has recently been reported by Lehnert (Inorg Chem 2019), and characterized by IR. The coupling of NO radicals without reduction, as suggested in the N_2O_2^* state, has an extremely long history in the literature; to my knowledge it is unknown except at very low temperature on noble metal surfaces. By contrast, NO oligomerization is well known upon reduction, e.g. reaction of NO^- with NO (eg, Lyman PNAS 2002), and this is the conventional understanding of NO coupling at FeN_4 sites, reduction first, then coupling to form N_2O and water (Lehnert reported a $\text{N}_2\text{O}_2^{2-}$ bridged between two FeN_4 centers, Inorg Chem 2014)

It is common to question the ability of DFT to model heterogeneous interface interactions, as well as water/proton H-bonding or H-atom transfer reactions. Therefore, additional verification is needed for those two intermediates. Conventional characterization of Fe-NO adducts is by IR or Raman spectroscopy, but such is difficult to use during catalytic turnover. The authors should consider generating the Fe(II)-NO adduct species (NO^*) by simply exposing to NO and then degassing, characterize the NO stretch at different potentials (comparable to literature values). Alternatively, do the same in anhydrous electrolyte solution, e.g. MeCN, and characterize the reduction of the Fe(II)-NO adduct species (NO^*) which should be reversible in absence of protons; compare reactivity with NO in the three different potential realms. This would verify the ability of Fe(II)-NO adduct species (NO^*) to couple w/ a second NO.

Therefore, I support the publication of this work but in a more specialized journal with inclusion of the characterization and descriptions in the supplemental. I also suggest a better description of the proposed mechanism regarding the Fe and NO oxidation states, referenced to the long history of this reactivity. The further experiments I suggest are not required for initial publication but would go a long way to address questions about their DFT simulations.

Reviewer #3 (Remarks to the Author):

The paper of Kim et al. demonstrates that hydroxylamine is produced with high faradaic efficiency from nitric oxide reduction on FeNC catalysts. They have used RRDE, OLEMS, XANES, DFT and single cell measurements. The experiments / calculations have been designed appropriately and the conclusions are supported by the data. The manuscript is well written and the methodologies are described sufficiently. Already the submitted version can be accepted. However, I recommend the following minor comments to be considered:

- I am a bit confused with the DFT calculations for two reasons: All steps from NO^* to NH_2OH^* are uphill in the energy diagrams, so the reaction should not happen at this potential. Also, the reaction coordinate, e.g. in Supplementary Fig. 17b, is not shown. NO^* to N_2O_2^* does not involve electrons, contrary to NO^* to NHO^* , but they are shown together. This representation is unusual, because every step does not correspond to one PCET.
- The Pt ring current was set to +1.4 V, because of the result from Supplementary Figure 10. But hydrazine (N_2H_4) can be oxidized on Pt. So NO reduction may yield hydrazine, which can be oxidized on the ring.
- It is known that N_2O gives an NO fragment at $m/z=30$, but in the supplementary figure 8 the $m/z=30$ signal is constant when the $m/z=44$ decreases from maximum intensity to the background (for example 500-700 seconds in c, or vice versa 1250-1500 seconds again in c).

Reviewer #1 (Remarks to the Author):

In “Selective electrochemical reduction of nitric oxide to hydroxylamine by atomically dispersed iron catalyst”, Kim et al. report a new single-atom Fe catalyst which is supported on graphene and stabilized with N atoms around the active site, mimicking the heme structure and is capable of reducing NO to H₂NOH with good selectivity alongside production of NH₃. The DFT calculations by SR and HK are based on an implicit solvation model coupled with the computational hydrogen electrode to construct reaction free energies diagrams and polarization curves using microkinetic analyses. It was found that the *N₂O₂ intermediate is rate-determining, and that the origin of super-Nernstian shift can be attributed to a stronger stabilization of the *NO intermediate at a more negative absolute potential. The high-quality calculations complement a thorough set of experimental analysis/characterization data and in my opinion warrants publication of this manuscript in Nature Communications after the following comments have been addressed.

Response: We thank the reviewer for the positive assessment and think that we have addressed all remaining points. Before going into the detail of the response, we would like to make an important comment of a change that was made in the notation. In reply to another reviewer’s comment, we denote the state NO* now as Fe^{II}-NO^{δ-}. This notation was chosen, to explicitly indicate the binding site of the adsorbate, as well as the oxidation state of Fe and the partial charge of the NO molecule under electrochemical conditions. The latter was found by a Bader charge analysis. This notation is now used throughout this response for consistency.

Detailed comments to be addressed prior to publication:

Main Text

1. Figure 3a: It would be helpful to overlay experimental polarization data for H₂NOH and NH₃ alongside the calculated curves, perhaps in a separate panel so that data are not overwhelming. The scale may be different however separate y axes can be used.

Response: We thank the reviewer for this remark and we agree that such a comparison is important. Figure R1 shows the experimental (total current) and theoretical (partial currents) polarisation curves at pH = 0 and pH = 3 (evaluated with the correct experimentally determined partial pressure of NO of 0.5 bar). Despite the qualitative agreement between theory and experiment, the theoretical lines show a shift of around 0.2 V compared to the experimental lines. In response to another reviewer’s comment, we additionally performed *in situ* IR experiments to confirm the key intermediates, and found that the NO adsorption signal was found below 0.2 V_{RHE}, while our microkinetic modeling predicts the NO adsorption below around 0 V_{RHE} (Figure R2). This indicates that our DFT energetics, even with using the HSE06 functional, underestimate the NO adsorption to the Fe centre by ca. 0.2 eV (which is in a reasonable error range of DFT energetics). Considering that the single-atom catalyst has a very localised electronic structure (e.g., *J. Phys. Chem. C* **2018**, *122*, 29307 or *ACS Catal.* **2020**, *10*, 7826), and the NO adsorbate has a strong radical character, a quantitatively accurate prediction of the NO adsorption energy is intrinsically difficult to describe with DFT functionals. More detailed discussions comparing RPBE, RPBE+U, and HSE06 results can be found from our response to the reviewer’s comment 3.

From the Figure R1, we also found that the scale of experimental and theoretical current density is different indicating that the Tafel slope in the theoretical data is underestimated. This could be due to several reasons, as a smaller charge-transfer coefficient, insufficient description of electrochemical reaction kinetics, or the exclusion of chemical reaction barriers which could contribute to the rate-limiting step and lower the potential dependence. Of particular, considering that our FeNC catalyst is a highly porous material, we believe the mass transport effects increase the Tafel slope and eventually could explain the early leveling off of the experimental data in comparison to the theoretical data (see Ringe *et al.*, *Nat. Commun.* **2020**, *11*, 1) which shows a constant increase until orders of magnitude higher

currents.

What we, however, want to note is that our model is not meant to be highly quantitative here. For this, a very long term work would need to be initiated including all the before mentioned effects. In this work, our purpose was to show that even with these approximations, our model can correctly describe that N_2O is formed at lower and $\text{NH}_2\text{OH}/\text{NH}_3$ at higher overpotentials. Most importantly, it also described the experimentally observed inverse pH trends on an RHE scale by the rate-limiting steps and charge dependence, and also the key intermediates proposed from theory were successfully captured from our additional *in situ* IR experiments. We consider this the most important qualitative statement of the paper, while quantitatively future studies should definitely work on the remaining issues.

Action: We have added the Figure R1 as Supplementary Figure 22. Detailed discussion has been also noted in Supplementary Note 4.

Figure R1. Comparison of experimental (total current, right y-axis) and theoretical (partial currents, left y-axis) polarisation curves at pH 0 and pH 3.

Figure R2. *In situ* ATR-SEIRAS analysis of FeNC-dry-0.5. **a,b**, Potential-dependent IR spectra from 0.8 to -0.2 V_{RHE} on FeNC-dry-0.5 in NO -saturated 1 mM $\text{HClO}_4/\text{H}_2\text{O}$ (**a**), and $\text{DCIO}_4/\text{D}_2\text{O}$ (**b**) electrolytes. All spectra were collected at a scan rate of 10 mV s^{-1} . **c**, Degree of rate control (DRC) analysis for the production of NH_2OH and N_2O from NORR using our micro-kinetic modeling and energetics calculated with the HSE06 functional.

2. Figure 3d: It is somewhat difficult to distinguish the orbital character around the *NO adsorbate. Can the authors provide a cartoon/schematic for what the orbital should look like relative to the graphene support, or otherwise outline more clearly the most relevant isosurface around *NO ?

Response: We agree with the reviewer that the plot was not that clear, in particular due to the overlapping carbon molecular orbitals. We have modified Figure 3 in the main manuscript and Supplementary Fig. 26 in the Supporting Information using the following two figures (Figures R3 and R4). In these, we minimised the size of the atoms to have a better focus on the isosurface, increased the iso-value to remove some of the carbon orbitals and also labeled the important atoms directly:

Action: We have modified Figure 3 in the main manuscript and Supplementary Fig. 26 in the Supporting Information using the following two figures (Figures R3 and R4).

Figure R3. Isosurface of the highest occupied molecular orbital (HOMO) of the bent Fe^{II}-NO^{δ-} state. The contour plot is given for the spin-up MO at the Γ -point at a wave function value of ± 0.004 (+ depicted by green, - depicted by blue).

Figure R4. Isosurface of the second highest occupied molecular orbital (2nd HOMO) of the linear Fe^{II}-NO^{δ-} state. The contour plot is given for the spin-up MO at the Γ -point and an isovalue of ± 0.004 (+ depicted by green, - depicted by blue).

Supplementary Information

3. "Current studies, however, also found that DFT+U cannot systematically correct all reaction intermediates towards more accurate hybrid functional energies, but rather also worsens some of the intermediate energies. Considering the high costs of hybrid functionals, we thus decided to stay with the RPBE functional."

This is very interesting and warrants further elaboration. Is the choice of $U = 2$ eV based on adsorption energies rather than the linear response method? Can the authors provide data on their tests with hybrid functionals, and how the different intermediates are affected by the application of U , i.e. how do these energies look with RPBE alone, RPBE+ U , and hybrid?

Response: We believe that the reviewer is raising an important point. DFT calculations based on the GGA functional have been shown before to be problematic for single-atom catalyst binding energies

with very localised electronic structure (e.g., *J. Phys. Chem. C* **2018**, *122*, 29307 or *ACS Catal.* **2020**, *10*, 7826). In response to the reviewer's request, we thus performed a full calculation of all reaction intermediate binding energies at zero surface charge on the RPBE+U2 (using $U = 2$ eV applied on the $l = 2$ orbitals of the Fe based on previous adsorption energy studies verified by comparison to HSE06 calculations and experimental data - *ACS Catal.* **2020**, *10*, 7826) and the HSE06 hybrid functional level. The following Table R1 shows in adsorption energies evaluated using the RPBE, RPBE+U2, and HSE06 functionals, the following Figures R5–7 show the resulting free energy diagrams on an RHE scale, Figure R8 shows the polarisation curves, and Figure R9 shows the degree of rate control (DRC) for the RPBE and HSE06 calculations.

From the energy shifts in Table R1, we first note that as expected the improved exchange description in both HSE06 and RPBE+U2 functional destabilises almost all adsorption states relative to the RPBE calculation. Later reaction intermediates are not critical as they are highly stabilised under the typically applied electrochemical potentials. However, the stronger destabilisation of the $\text{Fe}^{\text{II}}\text{-NHO}^{\delta-}$ state in the HSE06 calculations relative to the $\text{Fe}^{\text{II}}\text{-NHOH}^{\delta-}$ and $\text{Fe}^{\text{II}}\text{-NH}_2\text{OH}$ states has a direct consequence on the rate-limiting step of NH_2OH production. As seen in Figure R9, the rate-limiting step in the HSE06 calculations becomes the $\text{Fe}^{\text{II}}\text{-NO}^{\delta-}$ to $\text{Fe}^{\text{II}}\text{-NHO}^{\delta-}$ step over almost the full potential range, while at the RPBE level different states were rate-limiting depending on the potential. This reduces the super-Nernstian shift, which, however, still is around 60 mV for NH_3 and about the same value for most potentials for NH_2OH . Notable is also the reduction of the $\text{Fe}^{\text{II}}\text{-NO}^{\delta-}$ coverage at low overpotentials due to destabilisation of that state. This leads to an onset of NO adsorption at around 0 V_{RHE} close to the value that we observed during ATR-SEIRAS spectroscopy (Figure R2). N_2O production, in contrast, is limited by the $\text{Fe}^{\text{II}}\text{-N}_2\text{O}_2^{\delta-}$ formation step in both RPBE and HSE06 calculations, so that no pH dependence results on an SHE scale. As a result, the HSE06 calculations qualitatively reproduce the pH trends with the RPBE calculations, yet, with different shift values and rate-limiting steps.

In contrast to the HSE06 functional, the RPBE+U2 functional over-destabilises the $\text{Fe}^{\text{II}}\text{-NO}^{\delta-}$ state relative to the following states in the NH_2OH production pathway. This results in a rate-limiting step from $\text{Fe}^{\text{II}}\text{-NHO}^{\delta-}$ to $\text{Fe}^{\text{II}}\text{-NHOH}^{\delta-}$ in the potential range from -0.4 to $-0.2 V_{\text{RHE}}$ at pH 0 and $\text{Fe}^{\text{II}}\text{-NHO}^{\delta-}$ to mostly cover the surface in this region. This leads to a huge drop of the current with pH, much larger than expected from the experimental trends. Besides that, the $\text{Fe}^{\text{II}}\text{-N}_2\text{O}_2^{\delta-}$ intermediate is as destabilised as it is in the HSE06 calculation relative to the $\text{Fe}^{\text{II}}\text{-NO}^{\delta-}$ state. This makes the $\text{Fe}^{\text{II}}\text{-N}_2\text{O}_2^{\delta-}$ formation from $\text{Fe}^{\text{II}}\text{-NO}^{\delta-}$ feasible (see Figure R6) resulting in various steps being rate-limiting with Nernstian overpotential shifts and unrealistically high current densities (see Figure R8). Considering the resulting pH dependence of N_2O formation on an SHE scale together with the obtained independence from experimental data, we conclude that the RPBE+U2 calculation is not an optimal choice for this system. It might require a more careful optimisation of the U parameter and maybe even application to orbitals of the often radical adsorbate states to improve this description. For example, the NO adsorbate has a relatively strong radical character, and thus the electron localisation to the adsorbate is expected to be substantial. We, however, think that the HSE06 data reasonably proofs the validity of our conclusions and thus leaves it to future studies to optimise the RPBE+U formulation.

Action: We have added the Table R1 as Supplementary Table 2, the free energy diagrams as Supplementary Figs. 17–19, the polarisation curves as Supplementary Fig. 20 and also updated the rate-control analysis in Supplementary Fig. 19 with the HSE06 calculated results. We also added a corresponding short discussion of the differences between the functionals into the Supplementary Note 4, same to our response above to the referee.

Table R1. Adsorption energies of all intermediate states using the RPBE, RPBE+U2 and HSE06 functionals as obtained from the RPBE relaxed geometries at zero surface charge. The RPBE calculation was carried out with implicit solvation, using $\epsilon_b = 6$, while all other calculations were carried out under a vacuum environment. The most important changes in the binding energy from RPBE to HSE06 level of accuracy have been highlighted in red which have a direct consequence on the rate-limiting step (see text).

Adsorption state	ΔE	ΔE	ΔE
	(RPBE, eV)	(RPBE+U2, eV)	(HSE06, eV)
Fe ^{II} -NO ^{δ-}	-1.73	-0.81	-0.46
Fe ^{II} -NHO ^{δ-}	-1.66	-1.00	-0.37
Fe ^{II} -NHOH ^{δ-}	-1.55	-1.12	-0.94
Fe ^{II} -NH ₂ OH	-2.04	-1.85	-2.20
Fe ^{II} -N ₂ O ₂ ^{δ-}	-1.03	-1.17	-0.07
Fe ^{II} -HN ₂ O ₂ ^{δ-}	-1.77	-1.61	-1.47
Fe ^{II} -N ₂ O ^{δ-}	-3.38	-3.11	-2.17
Fe ^{II} -NH ₂ ^{δ-}	-3.59	-3.26	-3.31
Fe ^{II} -NH ₃	-4.75	-4.07	-4.54

Figure R5. Free energy diagram for NO reduction as calculated using the RPBE functional.

Figure R6. Free energy diagram for NO reduction as calculated using the RPBE+U2 functional.

Figure R7. Free energy diagram for NO reduction as calculated using the HSE06 functional.

Figure R8. Polarisation curves from micro-kinetic modeling based on adsorption energies calculated with the **a**, RPBE, **b**, HSE06, and **c**, RPBE+U2 functionals. The overpotential shift with pH is indicated by a gray arrow.

Figure R9. Degree of rate control analysis for the production of NH_2OH and N_2O from NORR as modeled using energies from the RPBE functional vs. energies from the HSE06 functional. The upper and central panel shows the DRC for NH_2OH and N_2O formation, respectively. The DRC indicates how strongly an intermediate influences the product formation rate (DRC = +1 (-1) means that stabilisation (destabilisation) of the intermediate enhances the reaction rate). All intermediates have been considered that at least partially in the considered potential range had a DRC > 0.1. The lower panel shows all non-zero coverages of intermediates over the whole potential window.

4. “(8) $\text{NH}_3^* + \text{H}^+ (\text{aq}) + \text{e}^- \rightleftharpoons [^-\text{NH}_3]^\ddagger \rightleftharpoons \text{NH}_3(\text{g})$ (chemical)” The proton-electron pair should not appear on the left side of this equation.

Action: We thank the reviewer for this important observation and have corrected this in the new version of the Supporting Information.

5. Supplementary Figure 15, I do not understand what the authors mean by “the structures were obtained from the zero surface charge calculations, despite for the ones where the additional access [excess] charge is annotated”. The authors mentioned previously that the geometry of $^*\text{N}_2\text{O}_2$ is actually optimized at a surface charge of -0.4 e or more negative as the adsorbate cannot be stabilized otherwise (page S6), so this caption appears to be inaccurate.

Response: The referee is correct, that $^*\text{N}_2\text{O}_2$ can be only stabilised at surface charges ≤ -0.4 e. In this picture, we show the geometry of each adsorbate with an applied excess charge which is closest to zero. For $^*\text{N}_2\text{O}_2$ this means we took the geometry that we obtained from the -0.4 e was taken, which is indicated by the charge inset in each figure.

Action: We have modified the caption to make this point clearer:

“Optimised unit cell structures of all intermediates. The structures were obtained from the zero surface charge calculations, despite some cases, which were not stable at this charge. For those cases, usually a more negative surface charge was required, and the figure shows the structure corresponding to the least negative charge (depicted by the inset) required to bind the adsorbate.”

6. In Supplementary Figure 16, it is seen that most of the intermediates along the NO reduction pathway respond in a similar way to surface charge, however, $^*\text{NH}_3$, $^*\text{NHO}$, and $^*\text{NHOH}$ are clear outliers. Can the authors provide a physical explanation for why these adsorbates have different energy dependences on applied surface charge?

Response: We thank the reviewer for this careful and important observation. It is true that there are some significant differences in the charge dependence between the adsorbates. In order to investigate this, we investigated in more detail the NH_3^* state which apparently shows the largest deviations from NO^* . From analysing the magnetic momentum, we first noted the presence of 2 unpaired electrons, one being localised at the Fe atom and the other one being delocalised over the carbon support. As shown in Figure R10, the HOMO is singly occupied by one of those electrons and formed by an antibonding overlap of the $\text{Fe}(d_{z^2})$ and the $\text{N}(p_z)$ orbitals. Adding more electrons to the NH_3^* state, thus destabilises the adsorbate state and leads to an inverse trend with surface charge compared to the NO^* state. In consequence, we can say that the surface charge dependence is a result of the character of the orbitals to be filled with the added excess charge.

Action: We have added the Figure R10 as Supplementary Fig. 28 and added the following sentence to the main manuscript:

“Additionally, it is important to mention that the presence of a finite surface charge does not always stabilise the adsorbate states (Supplementary Fig. 16). As an example, unlikely to the $\text{Fe}^{\text{II}}\text{-NO}^{\delta-}$ state, the $\text{Fe}^{\text{II}}\text{-NH}_3$ state has an anti-bonding, singly occupied (highest) MO (SOMO) (Supplementary Fig. 28). Filling of this SOMO results in a destabilisation at more negative potentials. This indicates that charge stabilisation is a complex function of the electronic structure of adsorption states, and thus a quantum-mechanical level of simulation reflecting the finite bias potential effect is a necessity for a comprehensive understanding of the electrochemical reaction mechanism.”

Figure R10. Isosurface of the singly occupied molecular orbital (SOMO) of the $\text{Fe}^{\text{II}}\text{-NH}_3$ state. The contour plot is given for the spin-up MO at the Γ -point at a wave function value of ± 0.004 (+ depicted by green, - depicted by blue).

7. Page S40, spelling error in equation “chrage”

Action: The mistake was corrected in the revised version of Supporting Information.

8. The computed polarization curves at pH 3 and pH 0, free energy diagram at -0.20 V vs. RHE, degree of rate-control analyses, HOMO-1 plot, and optimization of the NO^* intermediate as a function of surface charge are fine to publish in their present state.

Response: We thank the referee for the generally positive assessment again and hope that our changes have solved the remaining points.

Reviewer #2 (Remarks to the Author):

1. The report by Kim et al concerns the use of single-site FeN4 catalysts on graphene to investigate NO reduction reaction (NORR) pathways to N₂O and NH₂OH. The work is well done, extensive electrochemical analyses are supported by kinetic and DFT modeling; the catalytic materials were extensively characterized in several previous reports, (*point 1-1*) here only characterized by XANES to predict Fe oxidation states by applied potential (other characterizations are given in supplemental but not discussed: XPS, Mossbauer, Raman, EXAFS, TEM). (*point 1-2*) The results are presented in regard to the potential use of H₂-NO fuel cell reactor, which is fairly unique but not widely considered as viable.

Response: We are grateful to the reviewer for his/her constructive comments. Above, we separate the reviewer's comment 1 into '*point 1-1*' and '*point 1-2*' (highlighted by underlines). Because the *point 1-1* is interrelated with comment 2, they will be responded together in the comment 2. Regarding the *point 1-2*, we partly agree with the reviewer's comment that the H₂-NO fuel cell system is less attractive for the electricity generation as compared to the typical H₂-O₂ fuel cells.

However, we believe that the H₂-NO fuel cell system will be practically viable for the electrochemical production of value-added chemicals. NH₂OH, a main product from our H₂-NO fuel cell, is almost 10 times more expensive than that of NH₃. Considering even the NH₃ production from the electrochemical NO reduction can be market competitive based on the recent economic analysis (*ACS Energy Lett.* **2020**, *5*, 3647), our current study certainly promises the practical feasibility of electrochemical NH₂OH production.

Moreover, it is of note that the NORR is the key step, which determines the selectivity of the overall electrochemical denitrification process (Figure R11). Consequently, the current investigations of NO-to-NH₂OH conversion in device-level can provide an important stepping stone towards the technological development of electrochemical nitrate/nitrite denitrification, which is environmentally vital for balancing the natural nitrogen-cycle.

Action: Importance of the NO-to-NH₂OH conversion and its evaluation in the H₂-NO fuel cell system has been highlighted once again in the Conclusion section:

"the presented excellent performance metrics in device-level will provide an important stepping stone towards the technological development of fully sustainable electro-synthesis of valued nitrogen products from electrochemical nitrate/nitrite denitrification, which is environmentally vital for balancing the disturbed global nitrogen-cycle."

Figure R11. The electrochemical NO_x reduction reaction pathways, which show that NORR determines overall reaction selectivity toward N₂O, N₂, NH₂OH, or NH₃.

2. One issue is that this is the latest in a series of reports on the use of these FeN₄ catalysts (not entirely clean, but well characterized) to reduce H₂O₂ (Angewandte 2017) and O₂ (Energy and Environmental Sci 2018). Thus, the description of the catalyst formation and characterization in the text is limited, forcing new readers to do substantial background reading to understand the difference in materials used, e.g. FeNC-dry-1 vs FeNC-wet-1.

Action: In response to the reviewer's comments 1 (*point 1-1*) and 2, the following texts were newly introduced in the main manuscript to describe the catalyst formation and characterisation of all three catalysts in more detail:

"The labeling refers to homogenised condition (i.e., ball-milling of 'dry' precursor powders) and Fe content in the precursor mixture before pyrolysis at 1,323 K (see details in Methods section). As well-identified in our previous works,^{35,36} this catalyst is solely composed of isolated FeN_xC_y moieties (total Fe content ca. 1.5 wt%, no discernible Fe particles) conjugated on N-doped carbon substrate, as confirmed by a series of physical characterisation (see details in Supplementary Note 1 and Supplementary Figs. 1–5). Especially, ⁵⁷Fe Mössbauer spectroscopy and Fe K-edge extended X-ray absorption fine structure (EXAFS) reveal only two quadrupole doublets assigned to FeN_x sites and Fe-N(O) interaction in FeN_x sites, respectively, without any detectable spectroscopic signal from Fe clusters."

and

"The control catalysts were named 'FeNC-dry-1' and 'FeNC-wet-1', which were prepared as FeNC-dry-0.5 but with a two-fold higher Fe content in the precursor mixture and, for FeNC-wet-1, addition of a step for the aqueous complexation of Fe and phen, before milling the dried catalyst precursor (see details in Methods section).³⁵ A distinct property of the control catalysts compared to FeNC-dry-0.5 is the presence of metallic iron and Fe₃C (Supplementary Note 1). The quantitative analysis of their ⁵⁷Fe Mössbauer spectra identified that FeNC-dry-1 contains only ca. 0.2 wt% Fe particles and 2.8 wt% FeN_xC_y moieties while FeNC-wet-1 contains ca. 1.2 wt% Fe particles and 2.2 wt% FeN_xC_y moieties (Supplementary Table 1). Due to the ability of Fe particles to catalyse graphitization at the pyrolysis temperature, such Fe particles are surrounded by graphene shells (Supplementary Fig. 1), partially protecting them from immediate dissolution in acid medium."

3. Indeed, the report would be much enhanced by inclusion of Likewise, the electrochemical and product analyses are based on extensive previous work, e.g. series of papers by de Groot et al using rotating and ring-disc electrodes, which themselves followed numerous other investigations of the NORR pathways using porphyrins, phthalocyanines and heme enzymes/proteins. Thus the novelty and wide impact of the work may not be to the high standard of your journal.

Response: As noted by the reviewer, many previous efforts have been endeavored to investigate the NORR electrocatalysis on organometallic complexes such as metallo-porphyrins, phthalocyanines, and heme enzymes/proteins. Beyond the great advances in the molecular-level electrocatalysis, achieving the high operational stability is the next quest for the successful electrochemical denitrification of NO and other NO_x species. The present work directly aims to this issue, and the novelty of this work can thus be found in the following context.

We first achieved a prolonged NO-to-NH₂OH conversion in an electrochemical device, which has not been accomplished with molecular-level catalysts. Despite promising activity and selectivity of the organometallic complexes, these catalysts can be easily demetallised under the NORR conditions. In contrast, we demonstrated that the iron centre covalently embedded in the carbon matrix can catalyse the NORR stably even in highly corrosive conditions (even in 3M HClO₄) of electrochemical device equipped with polymer electrolyte membranes (PEMs).

Besides, we found an unprecedented super-Nernstian behaviour of NORR catalysed on the embedded iron centres. Our state-of-the-art electrochemical multiscale simulation elucidated that the super-

Nernstian behaviour is controlled by the local field developed in the electrical double layer region. In addition, the key intermediates of our proposing NORR mechanism based on computation were clearly supported by the new *in situ* attenuated total reflectance surface-enhanced infrared absorption spectroscopy (ATR-SEIRAS) study (will be discussed later in the reviewer's comment 7).

Therefore, considering great advances in the prolonged catalytic stability and the underlying fundamental insights, we, and also apparently the two other reviewers, believe that our findings will help tremendously the field of electrochemical NO reduction, which will be a milestone opening a new avenue for the viable electrochemical denitrification processes.

4. What is somewhat unique in this report, beyond the apparent selectivity for forming NH₂OH, is the electrochemical multiscale simulation studies and the resulting interpretation of the pathway intermediates. The authors use NO* to designate the Fe-bound intermediates, which is distracting and should be changed.

Response and Action: We apologise for any confusion due to the notation. In response to the comment, we changed the notation and intermediates are now denoted explicitly with reference to the Fe centre to which the molecules adsorb. In addition, we performed a detailed analysis of the partial charges, bond angles, distances and molecular orbitals which is described below the next point that the referee raised. As an example, the NO* intermediate is now depicted by the notation Fe^{II}-NO^{δ-}, indicating the partially negative charge of the NO molecule that we found under electrochemical reaction conditions and the oxidation state 2+ of the Fe centre which we found to be stable with respect to appliance of the electrochemical bias. Details are explained in the response to the next comment.

5. The schema in Fig 3 suggests that NO binds to the FeN₄ to generate NO*, but does not stipulate the Fe oxidation state! Both Fe(II) and Fe(III) states of FeN₄ moieties are known to bind NO, but with differing affinities and electronic structures. Likewise, structural and electronic variations between isoelectronic Fe(I)-(NO) and Fe(II)-(NO-) are known in different systems.

Response: We thank the reviewer for this valuable comment. As the reviewer pointed out, we agree that it is important to identify the oxidation state of Fe centre to understand its chemistry. Unlike organometallic compounds, however, we also would like to note that an isolation and subsequent full experimental characterisation of intermediate species is extremely difficult and mostly unavailable for heterogeneous electrocatalysts, as the metal centre is embedded into an extended system. Moreover, such an extended system is electronically connected to the potentiostat and additional electrons are supplied or drained by the potentiostat upon the given potential condition, it sometimes becomes ambiguous to assign an oxidation/charge state. Consequently, various *in situ* and *in operando* techniques as combined with theoretical modeling are the state-of-the-art methods for the most reliable understanding about the chemical natures of metal centres and adsorbates. This is what we are pursuing.

We first identified the oxidation state of the catalytically active Fe species as 2+ from *in situ* XANES measurement (Figure 2c in the main manuscript). Accordingly, in our DFT calculations, we modeled the Fe centre as 2+, and the NO was bound to Fe^{II}. When NO was bound to Fe^{II} under an open circuit condition, DFT results showed that N-O distance and Fe-N-O angle were 1.19 Å and 150°, respectively, supporting that the NO adsorbate is mostly neutral (N-O distance of free NO molecule is 1.16 Å). Upon negatively charging the system, that corresponds to the application of negative potential, the N-O distance increased until 1.25 Å and the Fe-N-O angle decreased until 125°, supporting the electronic occupation to N-O π* orbital which negatively charges the NO adsorbate (see Figure R12). *In operando* condition where a negative potential is applied, it is therefore reasonable to consider the NO adsorbate being negatively charged, but as the Bader partial charge below (Figure R12a) indicates, an assignment of full -1 charge to NO is also an oversimplification of the actual situation as the partial charge of NO being dependent on the applied potential and much smaller than 1.

Figure R12a further demonstrates that while charging the system, the partial charge of the Fe centre was found to marginally vary, implying that the oxidation state of the Fe centre remains as 2+. This corroborates the *in situ* XANES data, where no more reduction of Fe^{II} was observed.

By combining the results from *in situ* experiments and DFT, we conclude that the oxidation state of Fe is 2+ and the charge state of the NO adsorbate is in between 0 and -1 (denoted as δ^- hereafter) in the potential range that we are interested in. Similarly, we found all adsorbates to have negative Bader charges at negative surface charges, despite the NH₂OH* and NH₃* states (Figure R13). Thus, following the reviewer's kind suggestion, we revised our notations for designating Fe-bound intermediates to clearly denote their appropriate charge states and iron oxidation state, such as Fe^{II}-NO δ^- , Fe^{II}-NHOH δ^- , Fe^{II}-NH₂OH, Fe^{II}-N₂O₂ δ^- , and Fe^{II}-HN₂O₂ δ^- and the revised mechanism is shown in Figure R14.

This revised Figure R14 contains also the Fe^{II}-NHO δ^- intermediate which replaces the Fe^{II}-NHOH δ^- intermediate as the rate-limiting species following the more accurate hybrid functional DFT results which were obtained for this review.

Figure R12. Calculated properties for the Fe^{II}-NO δ^- state. **a**, Change of the Bader charges relative to the PZC of the NO adsorbate and the Fe centre in the optimised NO* state as a function of surface charge per unit cell. **b**, N-O bond distance and **c**, Fe-N-O angle in the optimised NO* state as a function of applied surface charge per unit cell.

Figure R13. Partial Bader charge of the adsorbates as a function of surface charge/unit area.

Figure R14. Reaction mechanism with the key rate-limiting intermediates as obtained from analysis of the microkinetic modeling results.

Action: We have added the Figures R12 and R13 as Supplementary Figs. 24 and 23, respectively, and also updated the reaction mechanism in Fig. 3b with Figure R14. We also have added the following discussion to the main manuscript:

“Using the electrochemical DFT approach and Bader charge analysis, we first found all adsorbates despite NH_3 and NH_2OH to exhibit a partially negative charge under negative electrode polarisation which we indicate by the superscript ‘ δ^- ’ (see Supplementary Fig. 23 for the Bader charge analysis).⁴³ In addition, the in situ XANES measurement (Fig. 2c) indicated an oxidation state of 2+ for the Fe centre. Bader charge calculations showed that the partial charge of the Fe centre marginally varies upon charging the surface (Supplementary Fig. 24). We thus conclude that Fe is always in the oxidation state 2+ throughout the reaction and denote e.g. the adsorption state of NO with $\text{Fe}^{\text{II}}\text{-NO}^{\delta^-}$.”

6. From my understanding of the voltammetry and simulations, the NO^* state corresponds to the $\text{Fe}(\text{II})\text{-NO}$ adduct which then can undergo two different paths, a 2 e-/H+ reduction to an $\text{Fe}(\text{II})\text{-NHOH}$ species or reaction with a second NO to form an $\text{Fe}(\text{II})\text{-N}_2\text{O}_2$ moiety. Both of these species are uncharacterized in experimental chemistry; a model $\text{Fe}(\text{II})\text{-NHOMe}$ species has recently been reported by Lehnert (Inorg Chem 2019), and characterized by IR. The coupling of NO radicals without reduction, as suggested in the N_2O_2^* state, has an extremely long history in the literature; to my knowledge it is unknown except at very low temperature on noble metal surfaces. By contrast, NO oligomerization is well known upon reduction, e.g. reaction of NO^- with NO (eg, Lyman PNAS 2002), and this is the conventional understanding of NO coupling at FeN_4 sites, reduction first, then coupling to form N_2O and water (Lehnert reported a $\text{N}_2\text{O}_2^{2-}$ bridged between two FeN_4 centers, Inorg Chem 2014).

Response: Thanks for the reviewer’s invaluable comment. As the reviewer pointed out and also stated in our previous response, NO is bound to Fe^{II} and also the NO adsorbate has a partially negative charge, i.e., $\text{Fe}^{\text{II}}\text{-NO}^{\delta^-}$. Thus, in our case too, an NO (partial) reduction accompanying an additional π^* occupation precedes the coupling of NO, and thus the conventional understanding about NO oligomerisation upon reduction can be applied to our mechanism. However, the NO reduction is not a single chemical event anymore, but the degree of NO reduction, in other words the extent of additional π^* occupation varies as a function of applied potential, since the N-O π^* orbital is hybridised with the carbon network (Figure R15) that is electronically connected with an electron bath-potentiostat.

Figure R15. Isosurface of the highest occupied molecular orbital (HOMO) of the bent $\text{Fe}^{\text{II}}\text{-NO}^{\delta-}$ state. The contour plot is given for the spin-up MO at the Γ -point at a wave function value of ± 0.004 (+ depicted by green, - depicted by blue).

In overall, we can summarise our mechanism of NO coupling as follows:

At the PZC, the Fe centre embedded in the carbon network has an oxidation state of 2+ at which the NO molecule can bind by hybridising N-O π^* orbital and Fe d_{z^2} orbital. As the system is being negatively charged at the finite potential, an additional electron provided from the potentiostat starts to occupy the $\pi^*(\text{NO})\text{-}d_{z^2}(\text{Fe})$ hybridised orbital (Figure R15), which partially reduces the NO adsorbate into $\text{NO}^{\delta-}$; a more negative potential applied to the system yields to a more reduced NO (Figure R12). In consistence with the previous understanding that NO^- can easily form an N-N bond with another NO molecule (*PNAS* **2002**, 99, 7340), NO coupling becomes energetically favourable only when the system is negatively charged and thereby the NO adsorbate has a partially reduced $\text{NO}^{\delta-}$ character (Figure R16).

Figure R16. Surface charge dependence of the $\text{Fe}^{\text{II}}\text{-NO}^{\delta-}$ and $\text{Fe}^{\text{II}}\text{-N}_2\text{O}_2^{\delta-}$ states. The energies were referenced to $\text{NO}_{(\text{g})}$, but without addition of the finite temperature corrections and ZPE.

Action: We also have added the following discussion to the main manuscript:

“At the potential of zero charge (PZC), we find the NO adsorbed to the iron centre is nearly neutral judging based on the Fe-N-O angle of 150° (Supplementary Fig. 24). In going to more negative potentials, however, the additional charge yields to a partially reduced NO (Supplementary Fig. 24). In consistence with the previous understanding that NO^- can easily form an N-N bond with another NO molecule,⁴⁵ NO coupling becomes energetically favourable when the system and consequently also the NO is more negatively charged.⁴⁶⁻⁴⁸ This leads to an inversion of relative energies of the $\text{Fe}^{\text{II}}\text{-NO}^{\delta-}$ state and the $\text{Fe}^{\text{II}}\text{-N}_2\text{O}_2^{\delta-}$ state (Supplementary Fig. 16).”

7. It is common to question the ability of DFT to model heterogeneous interface interactions, as well as water/proton H-bonding or H-atom transfer reactions. Therefore, additional verification is needed for those two intermediates. Conventional characterization of Fe-NO adducts is by IR or Raman spectroscopy, but such is difficult to use during catalytic turnover. The authors should consider generating the Fe(II)-NO adduct species (NO^*) by simply exposing to NO and then degassing, characterize the NO stretch at different potentials (comparable to literature values). Alternatively, do the same in anhydrous electrolyte solution, e.g. MeCN, and characterize the reduction of the Fe(II)-NO adduct species (NO^*) which should be reversible in absence of protons; compare reactivity with NO in the three different potential realms. This would verify the ability of Fe(II)-NO adduct species (NO^*) to couple w/ a second NO.

Response: In reading this excellent remark, we noticed that this would be a great idea to strengthen our conclusions. To identify key intermediates in NORR on the active Fe moieties, real-time ATR-SEIRAS has been employed here, which is particularly useful for electrocatalysis studies as its high surface sensitivity enables one to observe interfacial processes in the real-time reaction under the polarised conditions. The *in situ* ATR-SEIRAS results obtained at potentiodynamic condition revealed a pronounced peak at $1,691\text{ cm}^{-1}$ (Figure R17a). No considerable isotope shift of the peak was found when D_2O including DClO_4 was used in the electrolyte (Figure R17b), indicating that an adsorbate without H-atom is responsible for the peak. We calculated the harmonic vibrations using DFT/RPBE at zero applied surface charge (Table R2) and found only a single adsorbate to lie in this region, the $\text{Fe}^{\text{II}}\text{-NO}^{\delta-}$ state with a frequency of $1,702\text{ cm}^{-1}$. In addition, when looking at the potential-dependence, we found NO adsorption signal below $0.2\text{ V}_{\text{RHE}}$, which is comparable to our micro-kinetic modeling results (Figure R17c).

Besides the peak at $1,691\text{ cm}^{-1}$, the *in situ* ATR-SEIRAS results obtained at a constant potential of $-0.2\text{ V}_{\text{RHE}}$ further identified another peak at around $1,624\text{ cm}^{-1}$ (Figure R17d), which according to our calculations could be assigned in the accuracy trust region though to either of $\text{Fe}^{\text{II}}\text{-N}_2\text{O}_2^{\delta-}$ ($1,622.0\text{ cm}^{-1}$), $\text{Fe}^{\text{II}}\text{-NH}_2\text{OH}$ ($1,583.2\text{ cm}^{-1}$), or even $\text{Fe}^{\text{II}}\text{-NH}_3$ ($1,613.7\text{ cm}^{-1}$). However, in deuterated electrolyte condition, the peak remained at $1,624\text{ cm}^{-1}$ (Figure R17e), confirming that the peak results from neither $\text{Fe}^{\text{II}}\text{-NH}_2\text{OH}$ nor $\text{Fe}^{\text{II}}\text{-NH}_3$. In comparison with the theoretical data, we thus attribute the remaining peak without isotope shift to the $\text{Fe}^{\text{II}}\text{-N}_2\text{O}_2^{\delta-}$ state.

In conclusion, from the *in situ* ATR-SEIRAS spectroscopic studies, we could observe the proposed intermediate state which we claim to be the rate-limiting step in N_2O formation.

Table R2. Harmonic frequencies from DFT/RPBE in the region from $1,500$ to $2,000\text{ cm}^{-1}$ that has been experimentally studied with ATR-SEIRAS. Relaxed structures were taken at zero applied surface charge, despite the $\text{Fe}^{\text{II}}\text{-N}_2\text{O}_2^{\delta-}$ which was taken from the $q = -0.4\text{ e/unit cell}$ calculation and the $\text{Fe}^{\text{II}}\text{-NH}_2^{\delta-}$ state from the $q = -0.2\text{ e/unit cell}$ calculation.

Harmonic vibrational mode (cm^{-1})	Adsorbate state
1,502.7	$\text{Fe}^{\text{II}}\text{-NH}_2^{\delta-}$

1,622.0	$\text{Fe}^{\text{II}}\text{-N}_2\text{O}_2^{\delta-}$
1,583.2	$\text{Fe}^{\text{II}}\text{-NH}_2\text{OH}$
1,613.7	$\text{Fe}^{\text{II}}\text{-NH}_3$
1,631.5	$\text{Fe}^{\text{II}}\text{-NH}_3$
1,701.5	$\text{Fe}^{\text{II}}\text{-NO}^{\delta-}$

Figure R17. *In situ* ATR-SEIRAS analysis of FeNC-dry-0.5. **a,b**, Potential-dependent IR spectra from 0.8 to $-0.2 V_{\text{RHE}}$ on FeNC-dry-0.5 in NO-saturated 1 mM $\text{HClO}_4/\text{H}_2\text{O}$ (**a**), and $\text{DCIO}_4/\text{D}_2\text{O}$ (**b**) electrolytes. All spectra were collected at a scan rate of 10 mV s^{-1} . **c**, Degree of rate control (DRC) analysis for the production of NH_2OH and N_2O from NORR as obtained from micro-kinetic modeling using the HSE06 functional. **d,e**, Real-time IR spectra for NORR on FeNC-dry-0.5 at $-0.2 V_{\text{RHE}}$ in NO-saturated 1 mM $\text{HClO}_4/\text{H}_2\text{O}$ (**d**), and $\text{DCIO}_4/\text{D}_2\text{O}$ (**e**) electrolytes. **f**, Real-time profiles (with fitting lines) of integrated peak intensities of NO^* (1691 cm^{-1}) and N_2O_2^* (1624 cm^{-1}) in Figure R17e.

Action: We have modified the manuscript as follows:

- We have added the Figure R17 to the Fig. 3 in the main manuscript and to the Supplementary Fig. 25.
- We have added the Table R2 to the Supplementary Table 3.
- We have added this to the main text:

“To verify this mechanism, we performed detailed in situ attenuated total reflection-surface enhanced infrared absorption spectroscopy (ATR-SEIRAS) studies. By applying solvent isotope labeling, we identified non-hydrogen containing adsorbate peaks at $1,691$ and $1,624 \text{ cm}^{-1}$ at $-0.2 V_{\text{RHE}}$ (Fig. 3e–g).

These peaks are close to our theoretical RPBE-based estimate for NO stretching mode of the Fe^{II}-NO^{δ-} state at 1,702 cm⁻¹ and N₂O₂ stretching mode of the Fe^{II}-N₂O₂^{δ-} state at 1,622 cm⁻¹ (see Supplementary Table 3 for all calculated frequencies in the spectral region). We could thus respectively assign these peaks to the Fe^{II}-NO^{δ-} state and Fe^{II}-N₂O₂^{δ-} state and prove the existence of these intermediates along the reaction path. We also note that the potential dependent IR spectra showed NO adsorption signal below 0.2 V_{RHE} (Supplementary Fig. 25), which is comparable to our micro-kinetic modeling estimate (Supplementary Fig. 19)."

8. Therefore, I support the publication of this work but in a more specialized journal with inclusion of the characterization and descriptions in the supplemental. I also suggest a better description of the proposed mechanism regarding the Fe and NO oxidation states, referenced to the long history of this reactivity. The further experiments I suggest are not required for initial publication but would go a long way to address questions about their DFT simulations.

Response: We appreciate the critical comments from the reviewer. Since we very much believe in our work, we spent a considerable amount of effort to improve our conclusions and data. This comprises accurate and computationally elaborate hybrid functional calculations and state-of-the-art *in situ* ATR-SEIRAS spectroscopy. With this new data, we think that the conclusions are sufficiently validated, and we hope that the reviewer shares our enthusiasm about the novelty and impact into this work which makes it highly suitable for publication in Nature Communications.

Reviewer #3 (Remarks to the Author):

The paper of Kim et al. demonstrates that hydroxylamine is produced with high faradaic efficiency from nitric oxide reduction on FeNC catalysts. They have used RRDE, OLEMS, XANES, DFT and single cell measurements. The experiments / calculations have been designed appropriately and the conclusions are supported by the data. The manuscript is well written and the methodologies are described sufficiently. Already the submitted version can be accepted. However, I recommend the following minor comments to be considered:

Response: We thank the reviewer for this very positive assessment, and tried to address the minor points below.

1. I am a bit confused with the DFT calculations for two reasons: All steps from NO* to NH₂OH* are uphill in the energy diagrams, so the reaction should not happen at this potential.

Response: Indeed, all steps are uphill, as the referee correctly noted. However, we have to note that first of all, the free energy diagrams have not been corrected for an NO pressure different from unity and more importantly not for coverages. Including coverages in the free energy, every step in the free energy diagram is downhill and the overall thermodynamic driving force makes the reaction happening in the forward direction.

Action: We included the following sentence in the captions of the free energy diagrams to indicate that:

“The free energy diagram has been evaluated without accounting for pressures and coverages.”

2. Also, the reaction coordinate, e.g. in Supplementary Fig. 17b, is not shown. NO* to N₂O₂* does not involve electrons, contrary to NO* to NHO*, but they are shown together. This representation is unusual, because every step does not correspond to one PCET.

Response: The x-axis in this case is indeed not the number of PCET steps, but rather the reaction step. Due to the chemical steps involved in the reaction pathways, we found this representation to give the clearest picture, but we agree with the referee that this should be pointed out.

Action: In response, we have thus added the following sentence to all free energy diagram captions:

“The x-axis corresponds to the overall reaction coordinate being decomposed into the elementary steps (containing both PCET and chemical steps).”

3. The Pt ring current was set to +1.4 V, because of the result from Supplementary Figure 10. But hydrazine (N₂H₄) can be oxidized on Pt. So NO reduction may yield hydrazine, which can be oxidized on the ring.

Response: We are grateful to the reviewer for his/her constructive comment. As commented by the reviewer, hydrazine can be one possible product from the NORR, that we did not consider seriously in the original manuscript. Therefore, we first measured the hydrazine oxidation profile on the polycrystalline Pt ring electrode (Figure R18a). The result showed that the hydrazine oxidation starts at a potential below 0.3 V_{RHE} in a 0.1 M HClO₄ electrolyte and reaches a mass diffusion-controlled region at a potential above 0.6 V_{RHE}, indicating that the hydrazine oxidation reaction can occur at 1.4 V_{RHE} (a potential that we performed the RRDE investigations) and interfere the Pt ring current during the RRDE study.

Therefore, to identify whether the hydrazine is produced from NORR on the FeNC-dry-0.5 catalyst, we performed the RRDE studies again with a Pt ring potential of 0.8 V_{RHE}, at which the hydrazine can be oxidised but oxidations of NO, NH₃, and NH₂OH are silent. The new result revealed that no considerable oxidation current was monitored on the Pt ring electrode (Figure R18b), clearly indicating that hydrazine production is negligible during the NORR on the FeNC-dry-0.5. With this additional experiment, we thus

more clearly corroborated that main product of NORR catalysis on the FeNC-dry-0.5 is NH_2OH at high overpotential region, not NH_3 and N_2H_4 .

Action: In the revised manuscript, this information was newly added in Supplementary Fig. 10. The following text was also added in the revised manuscript:

“Here, NO-to- N_2H_4 conversion and its subsequent oxidation on the Pt ring could also be ruled out because of no considerable Pt ring current during the RRDE study performed at a Pt ring potential of 0.8 V_{RHE} , at which only N_2H_4 (not NO, NH_3 , and NH_2OH) can be oxidised (Supplementary Fig. 10).”

Figure R18. RRDE measurements. **a**, CV results of Pt ring electrode with various electrolyte conditions. Electrolytes are Ar-saturated 0.1 M HClO_4 solutions with and without 1.5 mM $\text{NH}_3/\text{NH}_2\text{OH}/\text{N}_2\text{H}_4$ and NO-saturated (ca. 1.3 mM) 0.1 M HClO_4 solution. The RRDE was rotated at 1,600 rpm. **b**, RRDE analysis during NORR on FeNC-dry-0.5 catalyst with ring electrode potential at 0.8 V_{RHE} . The black curve indicates the NORR polarisation curve of FeNC-dry-0.5 and red curve indicates the Pt ring current.

4. It is known that N_2O gives an NO fragment at $m/z=30$, but in the supplementary figure 8 the $m/z=30$ signal is constant when the $m/z=44$ decreases from maximum intensity to the background (for example 500-700 seconds in c, or vice versa 1250-1500 seconds again in c).

Response: In the original manuscript, we provided the online SFC/DEMS (or OLEMS) results without correction of the fragmentation because ion current of NO ($m/z = 30$) is much higher than that of N_2O ($m/z = 44$). For instance, in the Supplementary Fig. 8, the NO signal was shown after being reduced to 0.3 (or 0.5) times the original value. This would be a reason for the unclear change (or constant) in signal at $m/z = 30$ in spite of the decreased signal at $m/z = 44$.

However, the reviewer’s comment on the N_2O fragmentation to NO in the mass spectroscopy is definitely correct, and the fragmentation should be considered for better comparison. In the revised manuscript, we plotted the online SFC/DEMS results again, after correction of the initial signal at $m/z = 30$ by subtraction of 27% signal from $m/z = 44$ to remove N_2O contributions at $m/z = 30$ (Figure R19). This modification seems to slightly improve data quality (e.g., more clear mass-diffusion controlled region as highlighted with green rectangle in Figure R19). However, no considerable changes were made after the correction because initial signal at $m/z = 30$ overwhelms that induced by N_2O fragmentation.

Action: In spite of the no distinct difference before and after the correction, all the figures (Fig. 1, Supplementary Figs. 8, 9, and 14) in the revised manuscript were modified for accurate comparison, and this information was also noted in Methods section (copied below).

“The NO signal was shown after correction of the initial signal at $m/z = 30$ by subtraction of 27% signal

from $m/z = 44$ to remove N_2O contributions at $m/z = 30$.”

Figure R19. SFC/DEMS results of NORR electrocatalysis on FeNC-dry-0.5 catalyst measured at stepwise CA and CV conditions. The NO results was shown before (left) and after (right) correction of the initial signal at $m/z = 30$ by subtraction of 27% signal from $m/z = 44$ to remove N_2O contributions at $m/z = 30$.

Reviewer #1 (Remarks to the Author):

The Authors have more than thoroughly addressed my previous comments, and have also improved the manuscript both in response to Referees and of their own volition. In my opinion the manuscript is acceptable for publication in *Nature Communications*.

Reviewer #2 (Remarks to the Author):

I was satisfied with the authors responses and changes to the manuscript until I came to my question number 7 by the authors, which suggested they try to cleanly generate Fe(II)NO and the reduced Fe(II)NO anion and characterize by IR. They did indeed attempt the experiments using ATR-SEIRAs, but do not seem aware of how to interpret such experiments. A more thorough investigation is needed, with the expectation the significant changes in NO stretch would be observable with applied potential, as per their proposed mechanism. Please remove or amend the ATR discussion to rationalize comparison with previous small molecule literature.

Revision requested.

Reviewer #1 (Remarks to the Author):

The Authors have more than thoroughly addressed my previous comments, and have also improved the manuscript both in response to Referees and of their own volition. In my opinion the manuscript is acceptable for publication in Nature Communications.

Response: We thank the reviewer for his/her positive assessment of our work.

Reviewer #2 (Remarks to the Author):

I was satisfied with the authors responses and changes to the manuscript until I came to my question number 7 by the authors, which suggested they try to cleanly generate Fe(II)NO and the reduced Fe(II)NO- species and characterize by IR. They did indeed do the experiments using ATR-SEIRAs, but do not seem aware of how to interpret such experiments. A more thorough investigation is needed, with the expectation the significant changes in NO stretch would be observable with applied potential, as per their proposed mechanism. Please remove or amend the ATR discussion to rationalize comparison with previous small molecule literature.

There are many examples of vibrational spectra in the NOR literature assigned for NO⁺, NO, NO⁻ and HNO species in the same ligation. A one electron reduction causes the N-O stretch to decrease by >100 cm⁻¹, much more upon protonation. Below is nice compilation in table form from Lehnert [Speelman, Amy L., Corey J. White, Bo Zhang, E. Ercan Alp, Jiyong Zhao, Michael Hu, Carsten Krebs, James Penner-Hahn, and Nicolai Lehnert. "Non-heme high-spin {FeNO} 6–8 complexes: one ligand platform can do it all." *Journal of the American Chemical Society* 140, no. 36 (2018): 11341-11359.]

Their Figure R17 shows the data they used to make assignment of proposed intermediates.

First problem, they trust their deconvolution method to assign peaks specific to the Fe-N intermediates. Vibrational spectra are notoriously difficult to assign without using isotopic difference spectra. Standard practice is to use 14/15NO gas and generate difference spectra that will identify which vibrations involve the N atom.

Second problem, the suggested assignments in Table R2, below, which are confounding to a chemist with any knowledge in the area.

Vibrational spectroscopy of metal-bound nitrosyls are very informative, as they are very intense (high quantum probability)! The N-O bond stretch induces a large change in dipole moment which makes them easy to observe. This condition is dramatically changed if the N=O bond is lost or reduced, which is true of most of the species listed in Table R2!

Why even suggest that species such as FeII-NH₃ or FeII-NH₂OH would absorb in the same region? The N-H bond stretch should be in the 3000-4000 cm⁻¹ [Sickerman, Nathaniel S., Sonja M. Peterson, Joseph W. Ziller, and A. S. Borovik. "Synthesis, structure and reactivity of Fe II/III-NH₃ complexes bearing a tripodal sulfonamido ligand." *Chemical Communications* 50, no. 19 (2014): 2515-2517.]

Likewise, they report the Fe(II)-N₂O₂ species in the same range at 1622 cm⁻¹ but it should have two different N-O stretches if N-bound at a single Fe site. Again, 14/15N substitution is needed to even attempt to assign these peaks. N-N coupled species are rarer in single-molecule chemistry, [Xu, Nan, Adam LO Campbell, Douglas R. Powell, Jana Khandogin, and George B. Richter-Addo. "A stable hyponitrite-bridged iron porphyrin complex." *Journal of the American Chemical Society* 131, no. 7 (2009): 2460-2461.] but have been much discussed concerning the mechanism of enzymatic NOR. A good description of N₂O₂ and its redox congeners is found in this review, [Awasabisah, Dennis, and George B. Richter-Addo. "NO_x Linkage Isomerization in Metal Complexes." In *Advances in Inorganic Chemistry*, vol. 67, pp. 1-86. Academic Press, 2015.] as well as the table of vibrational frequencies for known

compounds below. Only the “free” N2O2 gasses appear in the region described.

The use of ATR to determine intermediates in Fe-NO reactions on solid surfaces has been used previously to investigate NO coupling [Lin, Rong, and Patrick J. Farmer. "O atom transfer from nitric oxide catalyzed by Fe (TPP)." *Journal of the American Chemical Society* 123, no. 6 (2001): 1143-1150. Kurtikyan, T. S., Martirosyan, G. G., Lorkovic', I. M., & Ford, P. C. (2002). Comparative IR Study of Nitric Oxide Reactions with Sublimed Layers of Iron (II)- and Ruthenium (II)- meso-Tetraphenylporphyrinates. *Journal of the American Chemical Society*, 124(34), 10124-10129.] There was controversy over the assignments, and suggestion that adventitious NO2 was involved in the observed reactions. The same would apply here.

Response: We appreciate the invaluable comments from the reviewer. We agree with the reviewer that the assignment of Fe-N intermediates would be highly speculative, as the issues regarding the dynamics of the kinetically linked intermediates during the electrocatalytic NO reduction process under an applied bias condition have not been fully addressed up to date. As the reviewer indicated, peaks should be carefully assigned based on the more extended works including isotope experiments. Hence, we tried to reduce the "speculative nature" as much as possible by removing the significant parts for Fe-intermediates, and rather focused on *in situ* monitoring of the (reduced) Fe^{II}-NO adduct species (in the reviewer's original question) under an applied bias. As the reviewer indicated, the significant changes in NO stretch were clearly observed as a function of an applied potential. We added attenuated total reflection-surface enhanced infrared absorption spectroscopy (ATR-SEIRAS) data at chronoamperometry polarisations between 0.8 and -0.2 V_{RHE} (see **Fig. R1**).

Fig. R1. **a,b**, *In situ* ATR-SEIRAS analysis of FeNC-dry-0.5 measured in NO-saturated 1 mM HClO₄+0.1 M KClO₄/H₂O (**a**) and 1 mM DClO₄ + 0.1 M KClO₄/D₂O (**b**) electrolytes. **c**, Integrated peak intensity of NO_{High} and NO_{Low} measured in the 1 mM DClO₄ + 0.1 M KClO₄/D₂O electrolyte. The IR spectra were collected at constant potentials of 0.4, 0.2, 0, and -0.2 V_{RHE} with a reference spectrum at 0.8 V_{RHE}.

In situ ATR-SEIRAS study identified two main bands at ca. 1723 (the high frequency NO; NO_{High}) and 1685 cm⁻¹ (the low frequency NO; NO_{Low}) at -0.2 V_{RHE} (**Fig. R1**). In addition, the positions of both bands were unchanged by solvent isotope labeling (in H₂O and D₂O solutions) (**Fig. R1a,b**), indicating that these bands are associated with nonprotonated species. IR bands of organometallic Fe-porphyrin complexes observed at ca. 1700 cm⁻¹ have usually been assigned to the Fe(η^1 -NO), where NO is bonded to Fe via nitrogen [JACS, 2000, 122, 7142; JACS 2001, 123, 1143]. Thus, we assigned these to the adsorbed NO species on the Fe center, *i.e.*, Fe^{II}-NO ^{δ^-} . Also, the high sensitivity of the NO vibration

frequency to the local chemical environments has been reported in previous studies [*Inorg. Chem.* 1999, 38, 100; *Inorg. Chem.* 2000, 39, 5102; *JACS* 2018, 140, 11341], and as also commented by the reviewer, the presence of two separate bands implies that there exist (at least) two chemically inequivalent Fe^{II}-NO^{δ-} species. However, these bands are separated by only ca. 40 cm⁻¹, and no appreciable signal was shown below 1600 cm⁻¹. Thus, the possibility of different binding modes such as η¹-ON and η²-NO could reasonably be excluded [Awasabisah, Dennis, and George B. Richter-Addo. "NO_x Linkage Isomerization in Metal Complexes." *In Advances in Inorganic Chemistry*, vol. 67, pp. 1-86. Academic Press, 2015]. Instead, slightly more reduced NO forming a more bent Fe-N-O geometry may explain the band at the lower frequency (NO_{Low}). Furthermore, the integrated peak intensities of both bands increased with decreasing an applied bias, inferring the increase of more Fe^{II}-NO^{δ-} species at lower potential (**Fig. R1c**).

At potentials below 0.2 V, both bands red-shift linearly with the lowering potential due to the Stark effect [*Electrochim. Acta*, 1996, 41, 623]. The slope of the shift is approximately ca. 50 cm⁻¹ V⁻¹, in accordance with previous reports [*J. Chem. Phys.* 1999, 111, 368; *Langmuir*, 2008, 24, 4352].

Fig. R2 a, *In situ* ATR-SEIRAS analysis of FeNC-dry-0.5 measured in a NO-saturated 1 mM HClO₄ + 0.1 M KClO₄/H₂O electrolyte. The IR spectra were collected at constant potentials of 0.4, 0.2, 0, and -0.2 V_{RHE} with a reference spectrum at 0.8 V_{RHE}. **b**, Potential and current profiles during the *in situ* ATR-SEIRAS analysis for (a). **c**, *In situ* ATR-SEIRAS analysis of Au thin film measured in a 1 mM HClO₄+0.1 M KClO₄ + 1 mM NH₂OH/H₂O electrolyte. The IR spectra were collected at CV from 0.8 to -0.2 V_{RHE}. **d**, Potential and current profiles during the *in situ* ATR-SEIRAS analysis for (c).

In addition, the N-H bond stretches in Fe^{II}-NH₂OH were clearly observed at ca. 2922 and 2855 cm⁻¹ (**Fig. R2a**), which corresponded to that of NH₂OH on Au film under an applied bias (**Fig. 2c**) [*J. Chem. Phys.* 2012, 137, 054714]. **Fig. R2b** represents the data collection point and current profiles during chronoamperometry polarisations between 0.8 and -0.2 V_{RHE}. In order to improve the signal to noise ratio in SEIRAS, we modified *in situ* ATR-SEIRAS experiment conditions: i) modifying the catalyst loading method on Au film, ii) using NO gas with the higher purity, and iii) increasing the ionic strength in electrolyte. First, we changed catalyst inks condition (1 mg catalyst + 6 μL Nafion solution (5 wt%) + 600 μL IPA). 200 μL of catalyst ink was sprayed on Au thin films coated on a Si prism. We used NO gas (99.9 %) and changed the electrolyte condition (1 mM HClO₄ with a 0.1 M KClO₄ salt) to increase ionic strength.

Action: We added Figures R1 and R2 in a main manuscript (Figure 3) and a Supplementary Information (Supplementary Figure 25), respectively. Detailed discussion was also amended in NORR mechanism section.